# A signal processing-based interpretation of the Nash-Sutcliffe efficiency

Le Duc[12], Yohei Sawada[1]

[1]Institute of Engineering Innovation, University of Tokyo, Tokyo, 113-8656, Japan
[2]Meteorological Research Institute, Tsukuba, 305-0052, Japan

*Correspondence to*: Le Duc (leduc@sogo.t.u-tokyo.ac.jp)

**Abstract.** The Nash-Sutcliffe efficiency (NSE) is a widely used score in hydrology but is not common in the other environmental sciences. One of the reasons for its unpopularity is that its scientific meaning is somehow unclear in the literature. This study attempts to establish a solid foundation for NSE from the viewpoint of signal progressing. Thus, a 10  simulation is viewed as a received signal containing a wanted signal (observations) contaminated by an unwanted signal (noise). This view underlines an important role of the error model between simulations and observations.

By assuming an additive error model, it is easy to point out that NSE is equivalent to an important quantity in signal processing: the signal-to-noise ratio. Moreover, NSE and the Kling-Gupta efficiency (KGE) are shown to be equivalent, at 15  least when there are no biases, in the sense that they measure the relative magnitude of the power of noise to the power of variation of observations. The scientific meaning of NSE suggests a natural way to define NSE=0 as the threshold for good/bad model distinction, and this has no relation with the benchmark simulation that is equal to the observed mean. Corresponding to NSE=0, the threshold of KGE is given approximately by 0.5.

In the general cases, when the additive error model is replaced by a mixed adaptive-multiplicative error model, the traditional NSE is shown to be prone to contradiction in model evaluations. Therefore, an extension of NSE is derived, which only requires to divide the traditional noise-to-signal ratio by the multiplicative bias. This has a practical implication: if the multiplicative bias is not considered, the traditional NSE and KGE underestimate/overestimate the generalized ones when the multiplicative bias is greater/smaller than one. In particular, the observed mean turns out to be the worst simulation 25  under the viewpoint of the generalized NSE.

## 1 Introduction

In hydrology, the Nash-Sutcliffe efficiency (NSE) is one of the most widely used similarity measures for calibration, model comparison, and verification (ASCE, 1993; Legates and McCabe, 1999; Moriasi et al., 2007; Pushpalatha et al., 2012; Todini and Biondini, 2017). However, Schaefli and Gupta (2007) pointed out a noticeable fact that NSE is not commonly used even 30  in environmental sciences despite the fact that calibration, model comparison and verification are also employed in such scientific fields. Does this mean that NSE is a special metric that is only relevant for hydrological processes? If this is not the

case, what causes this limited use outside of hydrology? One of the reasons can be traced back to the lack of consensual scientific meaning of NSE in literature.

NSE was firstly proposed by Nash and Sutcliffe (1970) by approaching calibration from a viewpoint of linear regression (Murphy et el., 1989)

$$NSE = 1 - \frac{\sum(o_i - s_i)^2}{\sum(o_i - \mu_o)^2} = 1 - \frac{\overline{(o-s)^2}}{\overline{(o-\mu_o)^2}}, \qquad (1)$$

where $s_i, o_i$ denote simulations and observations, respectively, $\overline{()}$ denotes the expectation, and $\mu_o = \bar{o}$ is the observed mean. The authors noted the analogy between NSE and the coefficient of determination $R^2$ in linear regression. Since $R^2$ measures

goodness-of-fit in linear regression, NSE should yield a similarity measure for our calibration problem. This use of $R^2$ implies that NSE regresses observations on simulations

$$o = as + b, \qquad (2)$$

where $a, b$ are the linear regression coefficients. Then the residual sum $\sum(o_i - s_i)^2$, which they called the residual variance, and the total sum $\sum(o_i - \mu_o)^2$, which they called the initial variance, are used in the definition of NSE. In general cases, the

residual sum should be $\sum(o_i - as_i - b)^2$. This points out that the underlying regression model implicitly assumes an unbiased regression line ($a = 1, b = 0$), which is rarely satisfied in reality.

A similar efficiency was introduced in Ding (1974) four year after the introduction of NSE

$$NDE = 1 - \frac{\sum(s_i - o_i)^2}{\sum(s_i - \mu_o)^2} = 1 - \frac{\overline{(s-o)^2}}{\overline{(s-\mu_o)^2}}, \qquad (3)$$

where we call this efficiency Nash-Ding efficiency (NDE). NDE can also be explained from the viewpoint of linear regression like NSE by switching the roles of $o, s$ in (2), i.e., regressing simulations on observations

$$s = ao + b. \qquad (4)$$

Under this regression equation, the coefficient of determination $R^2$ will take the form (3) if we again assume an unbiased regression line ($a = 1, b = 0$) as in the case of NSE. Note that in this case the total sum is given by $\sum(s_i - \bar{s})^2$, and because

of the no bias assumption $\bar{s} = \bar{o}$, this becomes $\sum(s_i - \mu_o)^2$ in (3). It is interesting to see that the hydrology community have preferred NSE in calibration, even though regression of observations on simulations does not show any advantage over regression of simulations on observations.

Identifying NSE to $R^2$ in linear regression was soon replaced by identifying NSE to skill scores in verification (ASCE, 1993;

Moriasi et al., 2007; Schaefli and Gupta, 2007; Ritter and Munoz-Carpena, 2013). Here a skill score measures a relative performance between a score and its benchmark or baseline (Murphy, 1988). This benchmark score is obtained by using a benchmark simulation, which is usually an easily accessible simulation that does not require complicated computation. The most common benchmarks are long-term or climatological means. Thus, applying to NSE, the nominator $\sum(o_i - f_i)^2$ is

simply the familiar mean squared error score (MSE), while the denominator $\sum(o_i - \mu_o)^2$ is now reinterpreted as the MSE of the benchmark given by the observed mean $f_i = \mu_o$. Equivalently, NSE can also be viewed as a normalized MSE with the normalizing factor $\sum(o_i - \mu_o)^2$ (Moriasi et al., 2007; Lamontagne et al., 2020).

However, when applied into forecast verification where simulations are replaced by forecasts, the special choice of $\mu_o$ as the benchmark does not conform with the purpose of using skill scores. Here the problem is that the observed mean can only be accessed after all observations are realized. This is not available at the time we issue forecasts, and therefore cannot be compared with our forecasts at that time. This subtle problem was noticed by several authors (Legates and McCabe, 1999; Seibert, 2001) and seasonal or climatological means were suggested as benchmarks instead of the observed mean. However, Legates and McCabe (2012) showed that the appropriate choice of benchmarks depends on hydrological regimes, leading to a more complicated use of NSE in verification. Therefore, they suggested to stick with the original NSE.

In recent years, starting with the work of Gupta et al. (2009), NSE has been recognized as a compromise of different criteria that measures an overall performance by combining different scores for means, variances and correlations. The decomposition form of NSE in terms of the correlation $\rho$, the ratio of standard deviations $\alpha = \sigma_s/\sigma_o$, and the ratio of means $\beta = \mu_s/\mu_o$ is given by

$$NSE = 2\alpha\rho - \alpha^2 - \frac{(\beta-1)^2}{(\sigma_o/\mu_o)^2}. \qquad (5)$$

Given this unintuitive form of NSE, Gupta et al. (2009) suggested a more intuitive score named the Kling-Gupta efficiency (KGE)

$$KGE = 1 - \sqrt{w_\rho(\rho - 1)^2 + w_\alpha(\alpha - 1)^2 + w_\beta(\beta - 1)^2}, \qquad (6)$$

where $w_\rho, w_\alpha, w_\beta$ are weights for individual scores and are usually set to 1. Note that mathematical form (6) is only one of many potential combinations of $\rho, \alpha, \beta$ that yield an appropriate verification score. In this multiple-criteria framework, the scientific meaning of KGE depends on the weights that we put to individual scores. However, unlike KGE, NSE defined by (5) is not a linear combination of the individual scores related to $\rho, \alpha, \beta$, therefore, the scientific meaning of NSE is even more obscure in this context. In other words, we can simply explain that NSE measures overall performances, but we cannot separate the contribution from each individual score.

One of weak points of the multiple-criteria viewpoint is that it explains the elegant form (1) by the unintuitive form (5). We suspect that there exists a more profound explanation for the elegant form (1) which also gives us the scientific meaning of NSE. In pursuing this explanation, we will come back to the insight of Nash and Sutcliffe (1970) when they first proposed NSE as a measure. This insight was expressed clearly in Moriasi et al. (2007) when they understood NSE as the relative

magnitude of variances of noise and variances of informative signals. This suggests us to approach NSE from the perspective of signal processing. We will show that NSE is indeed a well-known quantity in signal processing.

This paper is organized as follows. In Sect. 2, we revisit the traditional NSE from the viewpoint of signal processing on simulations and observations. Only with an additive error model imposing on simulations and observations, the nature and

behaviour of NSE in practice can be established. Since the additive error model implies variances of simulations are greater than variances of observations, in order to cover other cases Sect. 3 extends the error model in Sect. 2 by introducing multiplicative biases besides additive biases. Then an extension of NSE in these general cases is derived. Finally, Sect. 4 summarizes the main findings of this study and discusses some implications of using NSE in practice.

## 2 Specific cases: additive error models

### 2.1 The scientific meaning of NSE

From now on we will consider simulations and observations from the perspective of signal processing. According to this view, observations form a desired signal that we wish to faithfully reproduce whenever we run a model to simulate such observations. This simulation introduces another signal known as the received signal in signal processing, and is assumed to be the wanted signal (the observations) contaminated by a certain unwanted signal (noise). This means that we will have a

good simulation whenever model errors as represented by the noise are small. In this section, we assume a simple additive error model for simulations

$$s = o + b + \varepsilon, \quad (7)$$

where $b$ denotes constant systematic errors, and $\varepsilon \sim \mathcal{N}(0, \sigma_e^2)$ denotes random errors with the error variance $\sigma_e^2$. The two random variables $o$ and $\varepsilon$ are assumed to be uncorrelated.


Using the error model (7), it is easy to calculate two expectations in the formula of NSE

$$MSE = \overline{(s - o)^2} = b^2 + \sigma_e^2, \quad (8)$$

$$\overline{(o - \mu_o)^2} = \sigma_o^2, \quad (9)$$

leading to the following form of NSE

$$NSE = 1 - \frac{b^2 + \sigma_e^2}{\sigma_o^2}. \quad (10)$$

The reciprocal of the ratio $(b^2 + \sigma_e^2)/\sigma_o^2$ in (10) recalls the signal-to-noise ratio (SNR) in signal processing

$$SNR = \frac{P_{signal}}{P_{noise}} = \frac{\overline{o^2}}{\overline{(b+\varepsilon)^2}} = \frac{\overline{(\mu_o+o-\mu_o)^2}}{\overline{(b+\varepsilon)^2}} = \frac{\mu_o^2 + \sigma_o^2}{b^2 + \sigma_e^2}, \quad (11)$$

where $P_{signal}, P_{noise}$ are the power of the desired signal and noise, respectively. The greater SNR, the better the received signal.


In order to see the relationship between NSE and SNR, we notice that the error model (7) is preserved under the translations $(s, o) \rightarrow (s + \Delta, o + \Delta)$ where $\Delta$ is an arbitrary real number. This is easy to verify since the same error model is obtained when we add the same value $\Delta$ to $s, o$ on both sides of (7). A robust score should reflect this invariance, and therefore is required to be invariant under those translations. If this condition is not satisfied, we will get a different score every time we

change the base in calculating water levels for example. It is clear that NSE is translation-invariant while SNR is not. Indeed, we can easily increase SNR by simply increasing $\mu_o$

$$SNR(\Delta) = \frac{(\mu_o + \Delta)^2 + \sigma_o^2}{b^2 + \sigma_e^2}. \qquad (12)$$

This is because when a large $\Delta$ is added to the desired signal, its magnitude is almost dominated by $\Delta$ and the noise magnitude becomes negligible. This suggests that we can use the lower bound of $SNR(\Delta)$, i.e., the SNR in the worst case, as

a score to impose the translation-invariant condition

$$SNR_l = \frac{\sigma_o^2}{b^2 + \sigma_e^2}. \qquad (13)$$

This value is attained when $\Delta = -\mu_o$, which indicates the ratio of the power of variation $o - \mu_o$ to the power of noise. It is worth noting that the translational invariance is violated in the case of KGE since the ratio $\frac{\mu_s + \Delta}{\mu_o + \Delta}$ can vary considerably with $\Delta$.


Since the reciprocal of $SNR_l$ determines NSE in (10), it is more appropriate to define NSE in terms of the noise-to-signal ratio ($NSR = \frac{P_{noise}}{P_{signal}}$):

$$NSE = 1 - \frac{1}{SNR_l} = 1 - NSR_u, \qquad (14)$$

where we add the subscript u to NSR to emphasize that this is the upper bound of NSR corresponding to the lower bound of

SNR. Thus, under our additive error model, (14) points out that NSE is equivalent to the upper bound of NSR. More exactly, NSE measures the relative magnitude of the power of noise (the unwanted signal) and the power of variation of observations (the wanted signal with its mean removed). Similarly, it is easy to show that NDE (3) is also a simple function of $NSR_u$

$$NDE = 1 - \frac{b^2 + \sigma_e^2}{\sigma_o^2 + b^2 + \sigma_e^2} = \frac{\sigma_o^2}{\sigma_o^2 + b^2 + \sigma_e^2} = \frac{1}{1 + NSR_u}. (15)$$

Again, NSE and NDE are shown to be equivalent but this time from the perspective of signal processing.


This new interpretation of NSE has two important implications to the use of NSE in practice. Firstly, note that NSR depends not only on the power of noise but also on the power of signals in consideration. Thus, NSE should not be used as a performance measure in comparing two different kinds of signals. We may commit a possibly erroneous assessment like this: our model is better for the flow regime A than the flow regime B where this may be the consequence of the simple fact

that the signals in the case A are stronger than those in the case B. From its mathematical form, it is clear that NSR favours

high-power signals, i.e., strong signals always make NSR small, therefore, it is easy to get high NSE values for strong signals. Such NSE values may be wrongly identified as an indicator of good performances resulting to misleading evaluations on model performances.

Secondly, as a ratio of the power of noise to the power of variation of observations, $NSR_u$ suggests a natural way to define a threshold of NSE that divides simulations into good and bad ones. Note that $NSR_u = 0$ for perfect simulations, and increases with increasing the power of noise. At $NSR_u = 1$ noise has the same power as variation of the desired signal, and consequently corrupts the desired signal. In other words, we cannot distinguish variation of observations from noise, and model simulations therefore are useless. Corresponding to $NSR_u = 1$, we have the two thresholds NSE=0, and NDE=1/2. In

the context of skill scores, the NSE of zero is also chosen as the boundary between good and bad simulations by requiring that good simulations must have MSEs smaller than the MSE of the observed mean $s = \mu_o$. Clearly, the two interpretations are very different, even though they give the same benchmark NSE=0. Whereas the choice of the observed mean as the benchmark simulation is quite arbitrary in the latter, such a benchmark is not needed in the former. In fact, there exist many models yielding the value NSE=0, which are not necessarily the observed mean. A further argument supporting the former

approach is the failure of the latter approach when applied into the case of NDE. For the benchmark model $s = \mu_o$, NDE becomes $-\infty$, which means all other simulations are always better than this benchmark as measured by NDE.

**2.2 Random noise-to-signal ratio**

Recall that NSE is invariant under the translations along the vector $(1,1)^{\mathrm{T}}$ $(f, o) \rightarrow (f + \Delta, o + \Delta)$. However, for general translations $(f, o) \rightarrow (f + \Delta_f, o + \Delta_o)$ where the translation vector $(\Delta_f, \Delta_o)^{\mathrm{T}}$ is an arbitrary vector, NSE can take any value

$NSE = 1 - \frac{(b + \Delta_f - \Delta_o)^2 + \sigma_e^2}{\sigma_o^2}.$ (16)

Consequently, we can increase NSE simply by choosing appropriate $\Delta_f, \Delta_o$. In practice this approach is known as bias correction with the choice of $\Delta_f \approx -b, \Delta_o = 0$. Since NSE is not invariant under the general translations, misinterpretation on model performances can be easily committed. For example, let us consider two simulations: one with a systematic error and another with a random error

$s_1 = o + b,$     (17a)

$s_2 = o + \varepsilon,$     (17b)

where we assume $\sigma_o = b = \sigma_e$. Then both $s_1, s_2$ have $NSE_1 = NSE_2 = 0$, indicating that both simulations are corrupted by model errors. However, it is clear that two simulations are not equal. Modellers from their experience know that the first simulation is better since an almost perfect simulation can be easily obtained from $s_1$ just by subtracting from $s_1$ the bias

estimated from observations. In contrast, the performance of $s_2$ cannot be improved by any translation.

In order to avoid the misjudgement as above, it is desirable to have a score that is invariant under any translation. From (16), it is easy to see that the bias term causes NSE to vary with different displacements of $f, o$. This motivates us to decompose $NSR_u$ into two components:

$$NSR_u = \frac{b^2 + \sigma_{\tilde{e}}^2}{\sigma_o^2} = \frac{b^2}{\sigma_o^2} + \frac{\sigma_{\tilde{e}}^2}{\sigma_o^2} = SNSR_u + RNSR_u, \qquad (18)$$

where $SNSR_u$ denotes the systematic $NSR_u$ which changes with the general translations, and $RNSR_u$ denotes the random $NSR_u$ which keeps constant regardless of translations. Thus, $RNSR_u$ is an irreducible component of $NSR_u$ under any translation and acts as a lower bound of $NSR_u$. Similar to (14) we define a generally invariant version of NSE in terms of $RNSR_u$

$$NSE_u = 1 - \frac{\sigma_{\tilde{e}}^2}{\sigma_o^2} = 1 - RNSR_u. \qquad (19)$$

Here the subscript u is added to emphasize that this NSE is indeed the upper bound of the original NSE, i.e., the highest NSE can be reached just by translations. $NSE_u$ is identical to NSE when there are no biases in simulations. For the two simulations $s_1, s_2$ in (17a) and (17b), the new score yields $NSE_{u1} = 1, NSE_{u2} = 0$, which reflect our subjective evaluation. As we shall see shortly, $RNSR_u$ will help to ease our analysis on the behaviour of NSE considerably.


Similar to $NSE_u$, we define $NDE_u$ in terms of $RNSR_u$

$$NDE_u = \frac{1}{1 + RNSR_u}. \qquad (20)$$

We now prove an interesting fact that $NDE_u$ is indeed a more familiar quantity in statistics: the correlation coefficient $\rho$. This is easy to prove by making use of (7) in the definition of $\rho$

$$\rho = \frac{\overline{(s - \mu_s)(o - \mu_o)}}{\sigma_s \sigma_o} = \frac{\overline{(o - \mu_o + \varepsilon)(o - \mu_o)}}{\sigma_s \sigma_o} = \frac{\sigma_o^2}{\sigma_s \sigma_o} = \frac{\sigma_o}{\sigma_s} > 0, \qquad (21)$$

and in the definition of $\sigma_f^2$

$$\sigma_s^2 = \overline{(s - \mu_s)^2} = \overline{(o - \mu_o + \varepsilon)^2} = \sigma_o^2 + \sigma_{\tilde{e}}^2. \qquad (22)$$

By plugging (22) into (21), we obtain a one-to-one map between $\rho^2$ and $RNSR_u$

$$\rho^2 = \frac{\sigma_o^2}{\sigma_s^2} = \frac{\sigma_o^2}{\sigma_o^2 + \sigma_{\tilde{e}}^2} = \frac{1}{1 + RNSR_u} = NDE_u, \qquad (23)$$

which reveals a profound understanding on $\rho$, i.e., the correlation reflects noisiness under the error model (7). This is illustrated in Fig. 1 with the joint probability distributions of $s, o$ for different values of $\rho$. From (23), it is easy to find the lowest correlation at which a simulation is still considered to be good

$$NDE_u = \rho^2 \geq NDE \geq 1/2 \leftrightarrow \rho \geq \frac{1}{\sqrt{2}} \approx 0.7. \qquad (24)$$

It is worth noting that this critical value of $\rho$ is unknown in the literature.


## 2.3 Relationships between NSE, NDE, and KGE

In the previous section we have showed that the four variables $NSE_u$, $NDE_u$, $\rho$ and $RNSR_u$ are equivalent in the sense that they reflect noise levels in simulations. Since the correlation $\rho$ is a more popular variable with its support on the finite interval $[0,1]$, we will use $\rho$ as the main independent variable and view all scores as functions of $\rho$ in this section. Thus, expression of $NSE_u$ in terms of $\rho$ is given by

$$NSE_u = 1 - RNSR_u = 2 - \frac{1}{\rho^2}. \qquad (25)$$

Similarly, we disregard the contribution from the means $\mu_s, \mu_o$ to KGE in (6) and define its upper bound by setting all the weights to 1.0

$$KGE_u = 1 - \sqrt{(\rho - 1)^2 + \left(\frac{\sigma_s}{\sigma_o} - 1\right)^2} = 1 - \sqrt{(\rho - 1)^2 + \left(\frac{1}{\rho} - 1\right)^2}, \qquad (26)$$

where we have made use of (17) to get the last expression. Recall that although KGE is not invariant under the translations $(s, o) \to (s + \Delta, o + \Delta)$, by excluding the bias term, its upper bound $KGE_u$ now becomes invariant under any translation. It is usually accepted that NSE and KGE do not have a unique relationship, and therefore are not comparable (Konner et al., 2019). However, by focusing on their upper bounds, we can easily compare the two scores on the same plot as depicted in Fig. 2 which also plots NDE for completeness. Several important findings can be drawn from this figure.

Firstly, the three scores are monotonic functions of $\rho$. This is the consequence of the fact that their functional forms are one-to-one maps from $\rho$ to these scores. These bijections ensure that any score $NSE_u$, $NDE_u$ or $KGE_u$ can be used as an indirect measure of $RNSR_u$. In this sense, $NSE_u$, $NDE_u$, and $KGE_u$ are only different sides of the same $RNSR_u$, i.e., they are interchangeable in measuring noisiness in simulations. This points out that KGE has the same scientific meaning as NSE, which indicates the relative magnitude of the power of noise to the power of variation of observations. This fact has been demonstrated in several studies, e.g., Yassin et al. (2019). Although, KGE has been proposed in the multiple-criteria framework, it is interesting to see that the signal processing approach reveals its unexpectedly scientific meaning.

Since we can make any new score simply by assigning any monotonic function of $\rho$ to a score, we illustrate this process by rederiving NDE pretending that we do not know its mathematical form (4). For this purpose, we develop a new score from scratch called correlation efficiency (CE) by first defining its upper bound as

$$CE_u = \rho^2. \qquad (27)$$

Using (23), we rewrite (27) as

$$CE_u = \frac{1}{1 + RNSR_u}. \qquad (28)$$

Then replacing $RNSR_u$ by $NSR_u$, we reintroduce the bias term back into (28) and get the final version, which turns out to be NDE

$$CE = \frac{1}{1+NSR_u} = \frac{\sigma_o^2}{\sigma_o^2 + b^2 + \sigma_e^2} = NDE. \quad (29)$$

Similarly, we can deduce the translation-invariant form of KGE from (26) by writing $\rho$ in terms of $RNSR_u$, and then replace $RNSR_u$ by $NSR_u$

$$KGE = 1 - \sqrt{\left(\frac{1}{\sqrt{1+NSR_u}} - 1\right)^2 + \left(\sqrt{1+NSR_u} - 1\right)^2}. \qquad (30)$$

Recall that the original KGE (4) is not even invariant under the special translations $(s, o) \to (s + \Delta, o + \Delta)$. With the new KGE (30) the translational invariance is satisfied. However, replacing $RNSR_u$ by $NSR_u$ is not the only way to enforce the translational invariance, adding a new bias term such as $\frac{b^2}{\sigma_o^2}$ under the square root in (26) also works here.

Secondly, the choice of an appropriate score in practice can be determined by its magnitude and sensitivity. In this sense, Fig. 2 explains why modelers tend to favour KGE in practice. This is because $KGE_u$ is always greater than $NSE_u$, and at the same time $KGE_u$ is less sensitive to $\rho$ than $NSE_u$ since the derivatives of $KGE_u$ are always smaller than the derivatives of $NSE_u$. $NDE_u$ is also a good candidate in terms of the magnitude and sensitivity when $NDE_u$ is only slightly smaller than $KGE_u$.

Thirdly, the smaller the correlation, the more sensitive NSE and KGE. This is the consequence of the non-linear dependence of $RNSR_u$ on $\rho$ as expressed in (23). As a result, estimations of KGE and NSE are expected to have high uncertainties when correlations decrease. In contrast, NDE is less sensitive with decreasing $\rho$.

Finally, at the threshold $\rho = 1/\sqrt{2}$ the value of $KGE_u$ is approximately 0.5 (the exact value is $1 - \sqrt{\left(1/\sqrt{2} - 1\right)^2 + \left(\sqrt{2} - 1\right)^2}$), which is the lowest $KGE_u$ at which unbiased simulations are still considered to be good. It is also the lower bound for the modified KGE (30), which considers all simulations whether they are biased or not, due to the way it is constructed where $NSR_u \leq 1$ entails $KGE \geq 0.5$. For the traditional KGE (5), the lower bound for good simulations is not a well-defined concept since this KGE is not just determined by $NSR_u$. This threshold $KGE_*$, if exists, has to be equal or greater than 0.5 because otherwise we get a contradiction for unbiased simulations satisfying $KGE_* < KGE_u < 0.5$. As a result, we come to the necessary condition for a good simulation is that $KGE \geq 0.5$ for any form of KGE.

Similar to the threshold of $\rho = 1/\sqrt{2}$ for a good simulation, this threshold of KGE is unknown in the literature. In particular, this value is much greater than the corresponding threshold of $NSE_u$, which is zero. This relatively large gap can lead to misjudgement on model performances in practice since similar to NSE, modelers tend to consider KGE=0 as the threshold for good/bad model distinction (Anderson et al., 2017; Fowler et al., 2018; Siqueira et al., 2018; Sutanudjaja et al., 2018;

Towner et al., 2019). Thus, all models with KGE between [0,0.5] are wrongly classified to give good performances while they are indeed "bad" models. It is worth noting that Rogelis et al. (2016) assigned the value KGE=0.5 to be the threshold below which simulations are considered to be "poor".

The threshold KGE=0.5 is much larger than the KGE value calculated for the benchmark model when the simulation is equal to the observed mean, which is approximately -0.41 as shown in Knoben et al. (2019). They guessed that -0.41 is the lower bound of KGE for a good model. However, we have already seen that both the observed mean $s = \mu_o$ and the simulations with $NSR_u = 1$ agree on the same value NSE=0. How can we explain the different values of -0.41 and 0.5 in the case of

KGE? The reason is that the benchmark simulation does not follow the model error (7). It is clear that the regression line $s = o$ dictated by (7) is very different from the regression line $s = \mu_o$ in the case of the benchmark. Furthermore, (7) entails that $\sigma_s$ is always greater than $\sigma_o$ as shown in (22), which is not the case for $s = \mu_o$. As a result, the error model (7) cannot describe simulations with their variances smaller than observation variances, which is expected to commonly occur in practice. This raises the question whether the additive error model holds in reality. If this error model is not followed in

reality, can we still use NSE? Another important question is how we introduce the benchmark model $s = \mu_o$ into the framework developed so far to examine NSE and KGE? These problems require an extension of the error model (7) and will be further pursued in next section.

## 3 General cases: mixed additive-multiplicative error models

### 3.1 Validity of the traditional NSE

In order to extend the additive error model to the general cases, we first notice that the error model (7) indeed gives us the conditional distribution of simulations on observations. Since all information on simulations and observations is encapsulated in their joint probability distribution, we can seek the general form of this conditional distribution from their joint distribution in the general cases. For this purpose, we will assume that this joint probability distribution is a bivariate normal distribution

$$p\begin{pmatrix} s \\ o \end{pmatrix} = \mathcal{N}\left[\begin{pmatrix} \mu_s \\ \mu_o \end{pmatrix}, \begin{pmatrix} \sigma_s^2 & \rho\sigma_s\sigma_o \\ \rho\sigma_s\sigma_o & \sigma_o^2 \end{pmatrix}\right]. \qquad (31)$$

If the joint distribution is not Gaussian, we need to apply some suitable transformations to $s, o$ such as the root squared transformation $(s, o) \rightarrow (\sqrt{s}, \sqrt{o})$, the log transformation $(s, o) \rightarrow (\log(s), \log(o))$, the inverse transformation $(s, o) \rightarrow (1/s, 1/o)$, ... (Pushpalatha et al., 2012). When the joint distribution has the Gaussian form, the conditional distribution also has the Gaussian form (see Chapter 2 in Bishop (2006) for the proof)

$$p(s|o) = \mathcal{N}\left[\mu_s + \frac{\rho\sigma_s}{\sigma_o}(o - \mu_o), (1 - \rho^2)\sigma_s^2\right]. \qquad (32)$$

This implies the following form of the error model

$$s = \frac{\rho\sigma_s}{\sigma_o}o + \left(\mu_s - \frac{\rho\sigma_s}{\sigma_o}\mu_o\right) + \varepsilon = ao + b + \varepsilon, \qquad (33)$$

where $a = \frac{\rho\sigma_s}{\sigma_o}$, $b = \mu_s - \frac{\rho\sigma_s}{\sigma_o}\mu_o$, and $\varepsilon \sim \mathcal{N}(0, \sigma_e^2)$ with $\sigma_e^2 = (1 - \rho^2)\sigma_s^2$. In other words, simulations in the general cases contain both multiplicative and additive biases besides additive random errors. It is easy to verify that (7) is a special case of (33) when $a = 1$.

It is worth noticing that the nature and behaviour of NSE in Sect. 2 is constructed solely relying on the additive error model without any assumption on the joint probability distribution of $(s, o)$. Therefore, in this section, we again only assume that the error model is described by (33), i.e., a mixed additive-multiplicative error model. The joint distribution is no longer assumed to be a bivariate normal distribution, although (33) is derived from this assumption. This means that the marginal distribution of observations is not restricted to be Gaussian, and can be any probability distribution. However, two important identities obtained with the Gaussian assumption still hold

$$\rho = \frac{\overline{(s-\mu_s)(o-\mu_o)}}{\sigma_s\sigma_o} = \frac{\overline{(ao - a\mu_o + \varepsilon)(o-\mu_o)}}{\sigma_s\sigma_o} = \frac{a\sigma_o^2}{\sigma_s\sigma_o} = \frac{a\sigma_o}{\sigma_s}, \qquad (34)$$

$$\sigma_s^2 = \overline{(s-\mu_s)^2} = \overline{(ao - a\mu_o + \varepsilon)^2} = a^2\sigma_o^2 + \sigma_e^2 \rightarrow \sigma_e^2 = (1-\rho^2)\sigma_s^2. \qquad (35)$$

Can we now proceed by plugging the error model (33) to the formula (1) of NSE as in Sect. 2? The answer is definitely no because it makes no sense to plug (33) to (1) without first verifying the relevance of the traditional NSE under the error model (33). We now demonstrate the failure of the three traditional scores NSE, NDE, and KGE when they are applied outside of the additive error model through a simple example. Let us consider a model simulation with an additive random error

$$s_1 = o + \varepsilon, \qquad (36)$$

where we assume $\mu_{s1} = \mu_o = 0$, and $\sigma_o = \sigma_e$. This simulation indeed gives us the thresholds $NSE_1 = 0, NDE_1 = 0.5, KGE_1 = 0.5$ that distinguish good simulations from bad ones as we have examined in Sect. 2. It is very clear that we cannot improve this simulation since the power of random noise is equal to the power of observations. But this is not true if we measure performances with NSE, NDE, and KGE by constructing a new simulation which is half of $s_1$

$$s_2 = 0.5s_1 = 0.5o + 0.5\varepsilon. \qquad (37)$$

Calculating its NSE, NDE, and KGE, we obtain

$$NSE_2 = 1 - \frac{\overline{(o-s_2)^2}}{\sigma_o^2} = 1 - \frac{\overline{(0.5o - 0.5\varepsilon)^2}}{\sigma_o^2} = 1 - \frac{0.5\sigma_o^2}{\sigma_o^2} = 0.5, \qquad (38a)$$

$$NDE_2 = 1 - \frac{\overline{(s_2-o)^2}}{s_2^2} = 1 - \frac{\overline{(0.5\varepsilon - 0.5o)^2}}{\overline{(0.5o + 0.5\varepsilon)^2}} = 1 - \frac{0.5\sigma_o^2}{0.5\sigma_o^2} = 0., \qquad (38b)$$

$$KGE_2 = 1 - \sqrt{(\rho - 1)^2 + (\sigma_2/\sigma_o - 1)^2} = 1 - \sqrt{\left(1/\sqrt{2} - 1\right)^2 + \left(1/\sqrt{2} - 1\right)^2} = 2 - \sqrt{2} \approx 0.6. \qquad (38c)$$

Suddenly, NSE and KGE indicate that $s_2$ is better than $s_1$ considerably, although all we do is just halving $s_1$. In contrast, NDE gives a very different evaluation: $s_2$ is much worse than $s_1$. However, (37) in nature is equivalent to (36), and we should not make any simulation better or worse by just scaling the observations and the random error. This simple example is enough to show that the scientific meaning of the traditional scores like NSE becomes questionable when we introduce multiplicative biases into the error model. This can be traced back to a similar problem of the MSE score as demonstrated in Wang and Bovik (2009).

We show a further argument for the irrelevance of the traditional NSE under the error model (33) by proving that NSE (1) is not invariant under the translations that preserve the error model (33). In the case of the error model (7) we have shown that this additive error model is preserved under the translations $(s, o) \rightarrow (s + \Delta, o + \Delta)$. Geometrically, these translations move the joint distribution along the regression line $s = o + b$. In the general cases (33), the regression line becomes $s = ao + b$. This suggests that the error model (33) is preserved under the translations $(s, o) \rightarrow (s + a\Delta, o + \Delta)$, which indeed holds since

$$s + a\Delta = a(o + \Delta) + b + \varepsilon. \qquad (39)$$

When $a \neq 1$, these transformations cause NSE (1) to vary with $\Delta$, and therefore the traditional NSE is no longer a robust score under the error model (33).

### 3.2 An extension of the traditional NSE

In order to seek an appropriate form of NSE in the general cases, we rely on the nature and behaviour of the traditional NSE examined in Sect. 2 by imposing three conditions on the generalized NSE: (1) it measures the noise level in simulations; (2) it is invariant under the translations $(s, o) \rightarrow (s + a\Delta, o + \Delta)$; and (3) its random component, equivalently its upper bound, is invariant under all affine transformations $(s, o) \rightarrow (\alpha_s s + \Delta_s, \alpha_o o + \Delta_o)$, where $\alpha_s, \Delta_s, \alpha_o, \Delta_o$ are arbitrary real numbers. Note that we use affine transformations here due to the presence of both multiplicative and additive biases in the error model (33). We proceed by choosing a special transformation, i.e., the bias-corrected transformation $(s, o) \rightarrow ((s - b)/a, o)$. This results in an additive error model without biases

$$s_{bc} = \frac{s-b}{a} = o + \frac{\varepsilon}{a}, \qquad (40)$$

which suggests that we can define a new NSE in terms of the following upper bound of RNSR

$$RNSR_u = \frac{\sigma_e^2}{a^2 \sigma_o^2}. \qquad (41)$$

We now prove that (41) is indeed invariant under the transformations $(s, o) \rightarrow (\tilde{s} = \alpha_s s + \Delta_s, \tilde{o} = \alpha_o o + \Delta_o)$. In terms of $(\tilde{s}, \tilde{o})$, the error model (33) becomes

$$\tilde{s} = \alpha_s s + \Delta_s = \alpha_s \left( a \frac{\tilde{o} - \Delta_o}{\alpha_o} + b + \varepsilon \right) + \Delta_s = \frac{\alpha_s a}{\alpha_o} \tilde{o} + \alpha_s \left( b - \frac{a\Delta_o}{\alpha_o} \right) + \Delta_s + \alpha_s \varepsilon. \qquad (42)$$

Denoting $\tilde{a} = \alpha_s a / \alpha_o, \tilde{\varepsilon} = \alpha_s \varepsilon$, we recalculate (41) for the updated error model (42) with the updated parameters $\tilde{\sigma}_e^2 = \alpha_s^2 \sigma_e^2, \tilde{a}^2 = \alpha_s^2 a^2 / \alpha_o^2, \tilde{\sigma}_o^2 = \alpha_o^2 \sigma_o^2$

$$RNSR_u = \frac{\tilde{\sigma}_e^2}{\tilde{a}^2 \tilde{\sigma}_o^2} = \frac{\sigma_e^2}{a^2 \sigma_o^2}. \quad (43)$$

Thus, (41) is invariant under any affine transformation, which enables us to define the upper bound of the generalized NSE similar to (19)

$$NSE_u = 1 - RNSR_u = 1 - \frac{\sigma_e^2}{a^2 \sigma_o^2}. \quad (44)$$

This upper bound entails the desired form of the generalized NSE

$$NSE = 1 - \frac{b^2 + \sigma_e^2}{a^2 \sigma_o^2} = 1 - \overline{\left( \frac{o}{\sigma_o} - \frac{1}{\rho} \frac{s}{\sigma_s} \right)^2}, \quad (45)$$

where the last expression shows its practical form in comparison with the traditional form (1). We only need to check the invariant property of (45) under the translations $(s, o) \rightarrow (s + a\Delta, o + \Delta)$. Since these translations do not alter the bias term $b$, and are a subset of the affine transformations $(s, o) \rightarrow (\alpha_s s + \Delta_s, \alpha_o o + \Delta_o)$, they preserve (45).

In the introduction we have noticed that the decomposition form (5) of NSE is relatively unintuitive even though it is derived
from the elegant form (1). From Sect. 3.1 we know that (1) is indeed only relevant under the additive error model (7). It becomes irrelevant when multiplicative biases are introduced into (7). Therefore, if we continue to use the traditional NSE in the general cases, an unintuitive form of NSE will be expected as verified by (5). The appropriate NSE in such cases is the generalized NSE (45).

What is the scientific meaning of the generalized NSE (45)? Clearly, it measures the relative magnitude of the power of noise to the power of variation of observations when the multiplicative factor is removed. Thus, similar to the traditional NSE, the NSE value of zero still marks the threshold of good and bad simulations. It also attains the maximum equal to one when models do not have additive biases and random errors. However, there exists a subtle difference in the general cases: the perfect score $NSE = 1$ includes not only the perfect simulation $s = o$, but also all simulations with only multiplicative
biases $s = ao$. This means that this generalized score does not measure the impact of multiplicative biases. In evaluating model performances, therefore, we should consider both NSE (45) and the multiplicative factor $a$ although NSE should have a higher priority.

### 3.3 Behaviour of the generalized NSE

We now prove a surprising result: the upper bound of NSE in the general cases is the same as in the cases of the additive
error model, which is given by (25). By making use of the two identities (34), (35) on (44) we have

$$NSE_u = 1 - \frac{\sigma_e^2}{a^2 \sigma_o^2} = 1 - \frac{(1 - \rho^2) \sigma_f^2}{\rho^2 \sigma_f^2} = 2 - \frac{1}{\rho^2}. \quad (46)$$

Thus, in the general cases, correlations still reflect noisiness in simulations. This is illustrated again in Fig. 3 for the joint probability distributions of $s, o$ with the same $\rho = 0.9$ and different multiplicative factors $a$. From Fig. 3, it is seemingly counter-intuitive to realize that the noise levels are the same among all simulations given the same correlations 0.9. Clearly, all the points $(s, o)$ tend to spread wider when increasing $a$, which implies the noisiness increases. However, this misinterpretation results from our implicit assumption on the additive error model (7) for all the simulations, i.e., $a = 1$ for all the cases.

A further simple argument will show why this is the case that the noise levels are the same in Fig. 3. Let us consider a simulation $s = o + \varepsilon$. Thus, the simplest way to reduce the magnitude of the random error $\varepsilon$ is to multiply $s$ with a very small multiplicative factor $a$. By doing this, we have a new simulation $\tilde{s} = as$ with a new random error $\tilde{\varepsilon} = a\varepsilon$. Does this mean that $\tilde{s}$ is less noisy than $s$? Of course, this is not true at all since the noisiness is measured by the relative magnitude between the power of noise and the power of variation of observations, but not by the absolute magnitude of noise. When we multiply $s$ with $a$, at the same time we multiply $o$ with $a$, and as a result the relative magnitude is unaltered. This points out further that noisiness of all simulations $s = ao + a\varepsilon$ for any value of $a$ should be considered to be equivalent. The generalized NSE (45) just reflects this fact.

Since the upper bound of the generalized NSE is invariant when we introduce multiplicative biases into the additive error model (7), all conclusions in Sect. 2.3 still hold. Thus, it is legitimate to use the upper bounds of NDE and KGE expressed by (23) and (26), respectively, in the general cases. This implies that the values $NDE = 0.5$ and $KGE \approx 0.5$ remain to indicate the thresholds below which all simulations are considered to be "poor". The generalized NDE and KGE can be derived using the same procedure to obtain (29), (30) with the generalized $NSR_u = \frac{b^2 + \sigma_e^2}{a^2 \sigma_o^2}$ in place of the traditional $NSR_u = \frac{b^2 + \sigma_e^2}{\sigma_o^2}$. We derive the generalized NDE for illustration

$$NDE = \frac{1}{1 + NSR_u} = \frac{a^2 \sigma_o^2}{a^2 \sigma_o^2 + b^2 + \sigma_e^2}. \qquad (47)$$

It is worth noting that the form (26) of $KGE_u$ when rewritten using the error model (33), the variance term $\left(\frac{\sigma_s}{\sigma_o} - 1\right)^2$ will be replaced by $\left(\frac{\sigma_s}{a\sigma_o} - 1\right)^2$

$$KGE_u = 1 - \sqrt{(\rho - 1)^2 + \left(\frac{1}{\rho} - 1\right)^2} = 1 - \sqrt{(\rho - 1)^2 + \left(\frac{\sigma_s}{a\sigma_o} - 1\right)^2}. \qquad (48)$$

Combined with the generalized $NSE_u$ (46), we see that in practice if $a$ is not taken into account, i.e., the traditional NSE and KGE are still used, we underestimate/overestimate the generalized NSE and KGE when $a$ is smaller/greater than 1.

In order to check the work of the generalized versions of NSE, NDE, and KGE, we re-evaluate the performances of the two simulations (36), (37). In the case of $s_1$, since the multiplicative bias $a = 1$, the generalized efficiencies are identical to the traditional ones, therefore we still have $NSE_1 = 0, NDE_1 = 0.5, KGE_1 = 0.5$. Since $s_2$ does not have any additive bias, its generalized NSE, NDE, and KGE are identical to its corresponding upper bounds

$$NSE_2 = 1 - \frac{\sigma_e^2}{a^2\sigma_o^2} = 1 - \frac{\overline{(0.5\varepsilon)^2}}{0.5^2\sigma_o^2} = 1 - \frac{0.5^2\sigma_o^2}{0.5^2\sigma_o^2} = 0., \qquad (49a)$$

$$NDE_2 = \frac{a^2\sigma_o^2}{a^2\sigma_o^2 + \sigma_e^2} = \frac{0.5^2\sigma_o^2}{0.5^2\sigma_o^2 + (0.5\varepsilon)^2} = \frac{0.5\sigma_o^2}{0.5\sigma_o^2 + 0.5\sigma_o^2} = 0.5, (49b)$$

$$KGE_2 = 1 - \sqrt{(\rho - 1)^2 + \left(\frac{\sigma_2}{a\sigma_o} - 1\right)^2} = 1 - \sqrt{\left(\frac{0.5\sigma_o}{0.5\sqrt{2}\sigma_o} - 1\right)^2 + \left(\frac{0.5\sqrt{2}\sigma_o}{0.5\sigma_o} - 1\right)^2} \approx 0.5. \quad (49c)$$

Thus, we obtain the same results as $s_1$, which shows consistency between the two simulations as expected.

With the generalized NSE, it is now possible to deal with the benchmark model $s = \mu_o$. We exclude the trivial case $o = s = \mu_o$ and always assume $\sigma_o \neq 0$. This special simulation is equivalent to the following model error

$$s = 0 * o + \mu_o + 0, \qquad (50)$$

which implies $a = 0, b = \mu_o, \sigma_e = 0$ in (33). This specific error model highlights a problem that we have omitted when defining the generalized NSE: $RNSR_u$ (41), and therefore NSE (45), can only be defined for the cases $a \neq 0$. When $a = 0$,

simulations and observations are two uncorrelated signals ($\rho = 0$), and it makes no sense to state that the received signal (simulations) is the true signal (observations) contaminated by noise.

In order to assign an appropriate value of NSE for the cases $\rho = 0$, we rely on the continuity of $NSE_u$ with respect to $\rho$ as shown in (46). Let $\rho$ approach zero in (46), we get the limit $NSE_u = -\infty$. Since $NSE_u$ is the upper bound of NSE, this

entails $NSE = -\infty$. The same argument yields $NDE = 0, KGE = -\infty$ under the limit $\rho \to 0$. In other words, all simulations uncorrelated to observations, which include the observed mean, should be classified to the worst simulations with $NSE = -\infty$. It can be justified by noticing that information on variation of observations is totally unknown if only available is an uncorrelated simulation. The generalized NSE therefore provides a new interpretation of the benchmark simulation $s = \mu_o$. Rather than a benchmark marking the boundary between good and bad simulations, the observed mean is indeed the worst

simulation which can be beat by any simulations correlated to observations.

In order to clarify the above sophisticated problem, we summarize our arguments as follows:
- Under the perspective of signal processing, the additive error model cannot deal with the benchmark model $s = \mu_o$.
- In the additive error model, NSE=0 means that noise dominates informative signals, which is unrelated to the
observed mean.

- The mixed multiplicative-additive model enables us to interpret the case of the observed mean when the multiplicative bias a=0.

- However, the traditional NSE is not robust to multiplicative biases. When we design a new score robust to multiplicative biases, the observed mean should be interpreted as the worst simulation which gives us no information on observation variability.

- Although the observed mean can be easily obtained in hydrological model calibration and seems to be reasonable as a benchmark, it makes no sense to choose the observed mean as a benchmark simulation from the signal-processing viewpoint of NSE.

**4 Conclusion**

The Nash-Sutcliffe efficiency is a widely used score in hydrology but is not common in the other environmental sciences. One of the reasons for its unpopularity is that its scientific meaning is somehow unclear in the literature. There exist many attempts to establish a solid foundation for NSE from several viewpoints: linear regression, skill scores, multiple-criteria scores. This study contributes to these studies by approaching NSE from the viewpoint of signal progressing. Thus, a simulation is viewed as a received signal containing a wanted signal (observations) contaminated by an unwanted signal (noise). This view underlines an important role of the error model between simulations and observations, which is usually implicit in our assumption. Thus, our approach follows Bayesian inference in which an error model is formally defined first, then a goodness-of-fit measure is derived (Mantovan and Todini, 2006; Vrugt et al., 2008). The rational is to avoid the use of NSE as a predefined measure without an explicit error model like in generalized likelihood uncertainty estimation (Beven and Binley, 1992) which has caused a long debate in hydrology community (Mantovan and Todini, 2006; Stedinger et al., 2008).

By assuming an additive error model, it is easy to point out that NSE is equivalent to an important quantity in signal processing: the signal-to-noise ratio. More precisely, NSE measures the relative magnitude of the power of noise to the power of variation of observations. Therefore, NSE is a universal metrics that should be applicable in any scientific fields. However, due to its dependence on the power of variation of observations, NSE should not be used as a performance measure to compare different signals. Its scientific meaning suggests a natural way to choose NSE=0 as the threshold to distinguish between good and bad simulations in practice. This is because when NSE goes below zero, the power of noise starts dominating the power of variation of observations, meaning that noise distorts the desired signal and makes it difficult to extract the useful information. This choice has no relation with the interpretation that NSE=0 corresponds to the benchmark simulation equal to the observed mean, and all good simulations need be better than this benchmark.

Since NSE can be easily increased simply by adding appropriate constants to simulations and observations, we seek its upper bound $NSE_u$ under all such additions. $NSE_u$ is pointed out to correspond to the random component of NSR, and is a useful concept in analysing the behaviour of not only NSE, but also NDE and KGE. It turns out that $NSE_u$, $NDE_u$, and $KGE_u$ are different measures of noisiness, which can be mapped one-to-one between any two scores. More surprisingly, it is found that these scores in their turn can be expressed in terms of a more familiar quantity: the correlation coefficient. This implies that they do not introduce any new score, and can equivalently be replaced by $\rho$. In this sense, any new score can be constructed from $\rho$ with any monotonic function of $\rho$. This leads to an important finding: corresponding to NSE=0, we have NDE=0.5, $KGE \approx 0.5$ (not KGE=0), and $\rho \approx 0.7$ which mark the thresholds for good/bad model distinction. This has an importantly practical implication on the use of KGE since modelers usually identify KGE=0 for this threshold similar to NSE=0. Thus, models with KGE between 0-0.5 can be wrongly classified as good performances in practice.

Since the additive error model cannot describe the simulations that have variances smaller than observation variances, we need to work with a more general error model to deal with such cases. By assuming a bivariate normal distribution between simulations and observations, the general error model is found to be the mixed additive-multiplicative error model. Under the general cases, the traditional NSE is shown to be prone to contradictions where different evaluations on model performances can be drawn from a simulation by just scaling this simulation. Therefore, an extension of NSE need be derived. By requiring that the generalized NSE is invariant under affine transformations of simulations and observations induced by the general error model, which helps to avoid any contradiction, the most appropriate form is found to be the traditional one adjusted by the multiplicative bias. Again, this has a practical implication on the use of NSE and KGE: if the multiplicative factor is not taken into account and the traditional ones are used instead, both the scores are underestimated/overestimated when the multiplicative bias is greater/smaller than one. The threshold values NSE=0, NDE=0.5, $KGE \approx 0.5$, and $\rho \approx 0.7$ still hold with the generalized scores.

Finally, we summarize here some profound explanations that the signal processing approach to NSE proposes

- Despite their different forms, NSE, NDE, KGE and the correlation coefficient are equivalent, at least when there are no biases, in the sense that they measure the noise-to-signal ratio between the power of noise and the power of variation of observations.
- The threshold NSE=0 for good/bad model distinction follows naturally from the fact that at this value the power of noise starts dominating the power of variation of observations. The choice of a benchmark model like the observed mean required in the interpretation of such a threshold in the traditional approach is no longer needed in the context of signal processing.
- Furthermore, the signal processing-based approach seamlessly enables us to derive the corresponding thresholds for other scores like NDE and KGE in the same manner, a problem which is not well-defined if the benchmark

approach is still followed. Corresponding to NSE=0, the thresholds of KGE and the correlation coefficient are given approximately by 0.5 and 0.7, respectively.

- The traditional form of NSE only reflects the noise-to-signal ratio under the additive error model. It no longer reflects this when multiplicative biases are introduced, and as a result has an unintuitive form in the general cases.


- It is necessary to adjust the traditional NSE in the general cases to avoid potential contradictions in model evaluations. If the effect of multiplicative biases on the noise-to-signal ratio is not considered and the traditional one continues to be used, NSE is underestimated/overestimated when the multiplicative bias is greater/smaller than one.

- All simulations uncorrelated to observations are considered to be the worst simulation when measured by NSE or KGE because no information on variation of observations can be retrieved in these cases. The constant simulation

given by the observed mean $s = \mu_o$ belongs to this class of simulations. Therefore, in the view of signal processing the observed mean should not be used as a benchmark model.

*Author contribution.* LD raised the idea and prepared the manuscript. The idea has further been developed in discussion. YS corrected the treatment of NSE and KGE in hydrology and revised the manuscript.


*Competing interests.* The authors declare that they have no conflict of interest

*Acknowledgments.* This work was supported by the Ministry of Education, Culture, Sports, Science and Technology (MEXT) as the Program for Promoting Researches on the Supercomputer Fugaku, Large Ensemble Atmospheric and

Environmental Prediction for Disaster Prevention and Mitigation (hp200128, hp210166, hp220167), Foundation of River & basin Integrated Communications (FRICS), and JST Moonshot R&D project (grant no. JPMJMS2281).

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

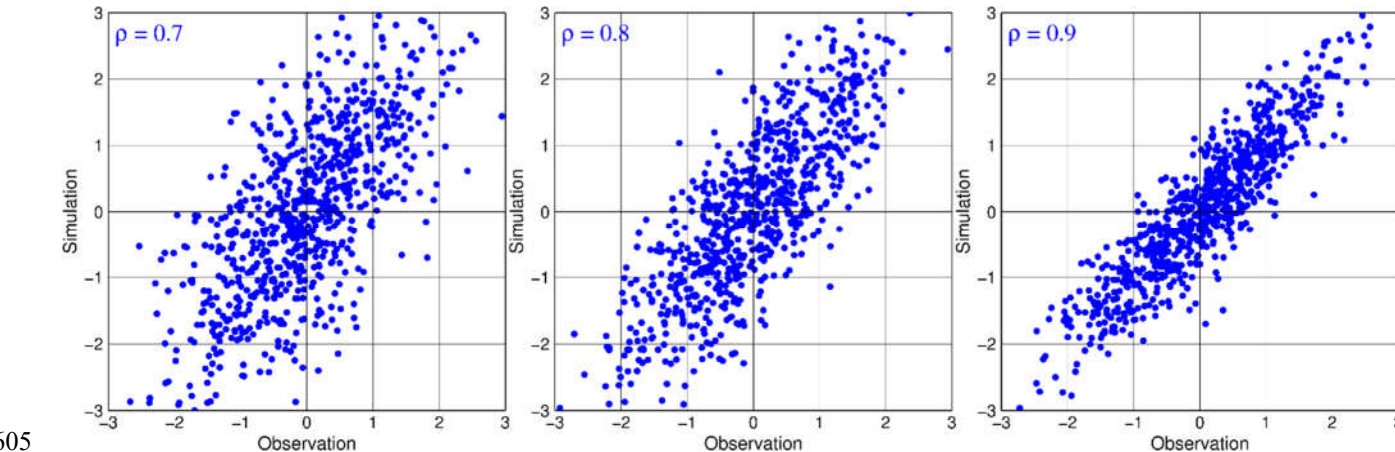


**Figure 1: Joint probability distributions of simulations and observations with different values of $\rho$ under the additive error model.**

Here we assume $b = 0$, $o \sim \mathcal{N}(0, 1)$, then the error model yields $\sigma_e = \sqrt{\frac{1}{\rho^2} - 1}\sigma_o$.

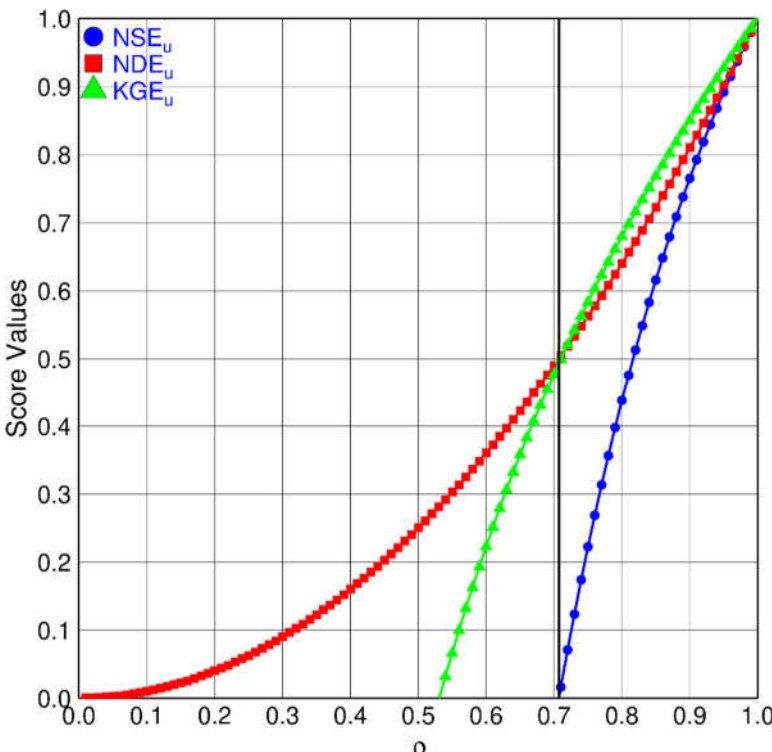

**Figure 2: The upper bounds of NSE, NDE and KGE as the functions of $\rho$. The solid line without symbols marks the boundary between bad simulations on the left, and good simulations on the right if simulations have no biases. If there exist biases in simulations, this boundary will shift to the right.**

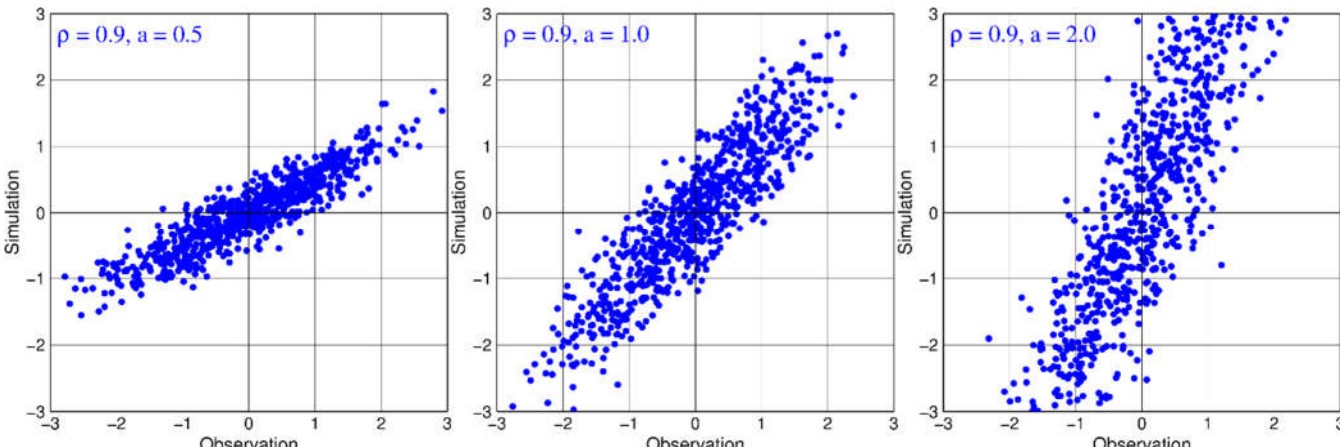

**Figure 3: Joint probability distributions of simulations and observations with the same $\rho = 0.9$ and different values of $a$ under the mixed additive-multiplicative error model. Here we assume $b = 0$, $o \sim \mathcal{N}(0, 1)$, then the error model yields $\sigma_e = \sqrt{\frac{1}{\rho^2} - 1} a \sigma_o$. The noise levels as measured by the generalized NSE indicate the same noisiness for all forecasts, even though the noisiness seemingly increases with increasing $a$.**