# Peer review of "A signal processing-based interpretation of the Nash-Sutcliffe efficiency"

_EGUsphere, 2022_

## Referee Comment (RC3)

[referee-annotated manuscript omitted]

---

## Community Comment (CC1)

**Origin of the NSE**

This is in response to the plea by the authors for insights on the popular Nash–Sutcliffe model efficiency criterion (NSE) - Lines 79-82.

I would like to share my memory of the origin of an alternate version of NSE. In hindsight, the scientific meaning of NSE boils down to what the term "initial variance" means. This was recently reported in Ding (2018) which may have escaped the attention of the authors.

Using the original notations, the NSE scale is recast from Equation (1) as follows:

$$R^2 = (F_0 - F)/F_0,$$

$$F_0 = \sum (o_i - \mu_o)^2,$$

$$F = \sum (f_i - o_i)^2,$$

in which: $R^2$ is the model efficiency, i.e. NSE; $F_0$ is the "initial variance" of the observations, $o_i$; $F$ is the "residual variance" of the forecasts, $f_i$; subscript $i$ is time; and $\mu_o$ is the mean of the observations.

In the literature, there existed an alternate definition presented by Ding (1974) of the so-called "initial variance", $F_0$. The alternative, symbolized by, say, $F_{0-d}$, appeared four years after Nash and Sutcliffe (1970), not year 1974 as written in the preprint.

I defined $F_{0-d}$ directly from $F$ by replacing the observations by the mean, i.e.

$$F_{0-d} = \sum (f_i - \mu_o)^2,$$

$F_{0-d}$ thus becomes the sum of squares of the error (SSE) of a model measured against the observed mean flow. This is different from $F_0$ which is simply the variance of the observations.

The origin of an observed-mean-flow benchmark or baseline was embedded as $\mu_o$ in the "initial variance" of both standard and alternate versions. But the alternate version, $F_{0-d}$, shows unambiguously the observed mean flow, a baseline.

The alternative was preserved in Ding (1974, Eqs. (40) & (47)) where the criterion or scale was called, after Nash (1968-69), the model efficiency, $R^2$, same in notation as the coefficient of determination (Line 39). Absent from the paper was the name of Sutcliffe.

Nash was the Journal of Hydrology editor of the 1974 paper of mine, and, as far as I remember, had kept silent on the alternate definition of the "initial variance", a technical term he first coined. In Ding (1974) paper, the raw source, Nash (1968-69), was cited, but not the subsequent one, Nash and Sutcliffe (1970). The alternate efficiency scale could be named after both Nash and Ding as NDE (1974) to differentiate it from the standard NSE (1970) after Nash and Sutcliffe.

Thanks for the opportunity to reflect on the genesis of a classical but imperfect metric in evaluating the performance of hydrologic models. So rudimentary is the observed-mean-flow baseline, and so imprecise the Nash–Sutcliffe efficiency (NSE) scale. But alternatives exist to both the baseline, e.g., Ding (2018), and the standard scale, e.g., Ding (1974). A combination of the two alternatives will lead to an invention of a new metric, "elegant" and "intuitive", both characterizations (Line 79) favoured by the authors of a metric.

References

Ding, J. (2018). "Interactive comment on "On the choice of calibration metrics for "high flow" estimation using hydrologic models" by Naoki Mizukami et al." doi: https://doi.org/10.5194/hess-2018-391-SC1.

Ding, J. Y. (1974). "Variable unit hydrograph". Journal of Hydrology, 22.1-2, pp. 53-69.

---

## Author Comment (AC1)

**Reply to Dr. Ding's comment**

We would like to thank Dr. Ding for the sharing on the interesting history of developing the Nash-Sutcliffe efficiency (NSE) and a related variants which was proposed but has never been considered in practice. Here are our explanations for this variant:

The following efficiency was introduced in Ding (1974) four year after the introduction of NSE in Nash and Sutcliffe (1970)

$$NDE = 1 - \frac{\sum(f_i - o_i)^2}{\sum(f_i - \mu_o)^2} = 1 - \frac{\overline{(f-o)^2}}{\overline{(f-\mu_o)^2}}, \quad (1)$$

where we call this efficiency NDE as in your comment. This means that instead of regressing observations on forecasts as in the case of NSE $o = f + \varepsilon$, NDE regresses forecasts on observations $f = o + \varepsilon$ and takes the resulting coefficient of determination as a similarity measure.

Under the additive error model

$$f = o + b + \varepsilon, \quad (2)$$

NDE becomes

$$NDE = 1 - \frac{b^2 + \sigma_e^2}{\sigma_o^2 + b^2 + \sigma_e^2} = \frac{\sigma_o^2}{\sigma_o^2 + b^2 + \sigma_e^2}. \quad (3)$$

This score is surprisingly the square of the correlation efficiency CE that we show in Equation 25 in the manuscript. We derived CE to illustrate that any score can be constructed from a monotonic function of the correlation coefficient $\rho$, and did not know that CE was indeed proposed in the literature. In the revised manuscript we will add a citation to your study for this efficiency.

Thus, NDE is equivalent to NSE, and both measure the noise-to-signal ratio in forecasts. However, there is a potential problem here if NDE was chosen instead of NSE in the history. From (1) like NSE we may wrongly conclude that NDE=0 marks the boundary between skillful and unskillful forecasts. But from (3) this threshold of NDE should be NDE=1/2 when the power of noise starts dominating the power of variation of observations $\sigma_o^2 = b^2 + \sigma_e^2$. Note that in the case of NSE, both equations similar to (1) and (3) yield the same threshold NSE=0.

Reference

Ding, J. Y.: Variable unit hydrograph. *Journal of Hydrology*, **22.1-2**, 53-69, 1970.

Nash, J. E., and Sutcliffe, J. V.: River flow forecasting through conceptual models. Part 1: A discussion of principles. *Journal of Hydrology*, **10(3)**, 282–290, doi:10.1016/0022-1694(70)90255-6, 1970.

---

## Author Comment (AC2)

We would like to thank the reviewer for your constructive comments to the manuscript. Here are our explanations for the unclear points that the reviewer concerned. But at first, we would like to reproduce the reviewer's comments in a black font:

The paper deals with a novel interpretation of the NSE score measure starting from the observation that it is mostly used in hydrology and poorly exploited in other sciences.

This interpretation is based on a signal processing viewpoint.

While the paper is interesting and useful the following concerns are at the hand.

The NSE interpretation provided is based on a model error for forecast which is given in eq 5 and is basically driven by a Gaussian random error being it equal to noise in the signal processing viewpoint. This basic foundation provides by itself a lack of generality with respect to application of NSE to other sciences included hydrology. In hydrology the NSE is intended as a model performance metric where the difference between model and observation is not limited to noise. Differences between model and observation but also differences between observation and reality and also differences between different models can be analyzed by means of NSE or KGE whose meaning is quite clear and leaves no doubts in my personal opinion.

In a hydrologic model, but also in other earth sciences, different models may arise because different processes are modelled in a stochastic or determinist way and/or because some processes are described or neglected following the fact that they can be more or less important according to the time-space scale of application and modeling purpose. Hence the difference between model output and observations may be very different from what given in eq 5, It can be deterministic or stochastic, and affected by deterministic or stochastic (or both) variability.

As a consequence, it seems that the proposed analysis, while interesting and well founded in the context of the signal processing field (or any other fields where only noise provides difference between model output and observation). In the same light one may not accept the "general case" version of NSE which is obtained by considering the multiplicative error, beside the additive error, defined in eqs (32). Even in this case the "general case" should be addressed as relative to which field of application, besides the field of signal processes or affine methods.

I believe the authors should strongly address this issue in a revised version of the manuscript.

Reply: First of all, we totally agree with the reviewer on the point that the additive error model that we consider in Section 2 is relatively limited and cannot cover all complicated relationships

between models and observations in reality. Our purpose when using this additive error model is to show that the scientific meaning of NSE is truly revealed under this error model, which turns out to be the well-known signal-to-noise ratio in signal processing.

Due to the limit of the additive error model, in Section 3 we consider more general cases which can be described by joint probability distributions of model simulations and observations. Although the reviewer mentioned that general cases have not been dealt, we believe that our mixed multiplicative-additive error model addressed a wide range of cases of model-observation fitting. We would like to emphasize here that any relationship between model simulations and observations can always be analyzed through their joint probability distributions. Then we assume that such distributions are bi-variate normal distributions, which is usually observed in reality. Again, we would like to emphasize that we do not assume the mixed multiplicative-additive error model for the general cases. Rather than that, this multiplicative-additive error model is a consequence of the bi-variate normal distribution of model simulations and observations.

Of course, if this joint distribution is not Gaussian, we cannot expect to see the mixed multiplicative-additive error model and the relationship between model simulations and observations may be even more complicated as the reviewer commented. However, we believe that there are some suitable transformations to make observations and forecasts follow Gaussian distribution in most cases including hydrological applications. The generality of our error model described above has already been discussed in the current manuscript:
Since all information on forecasts and observations is encapsulated in their joint probability distribution, we can seek the general form of this conditional distribution from their joint distribution in the general cases. For this purpose, we will assume that this joint probability distribution is a bivariate normal distribution. If the joint distribution is not Gaussian, we need to apply some suitable transformations to $f, o$ such as the root squared transformation $(f, o) \rightarrow (\sqrt{f}, \sqrt{o})$, the log transformation $(f, o) \rightarrow (\log(f), \log(o))$, the inverse transformation $(f, o) \rightarrow (1/f, 1/o)$, ... (Pushpalatha et al., 2012).

We have shared the same opinion with the reviewer that the meanings of NSE and KGE "is quite clear and leaves no doubts" until we have come up with a counterexample based on the mixed multiplicative-additive error model. Let us consider a model simulation with a random error
$$s_1 = o + \varepsilon, \qquad (1)$$
where we assume $\mu_{s1} = \mu_o = 0$, and $\sigma_o = \sigma_e$. This simulation in deed gives us the boundary between "good" and "bad" simulations from the viewpoint of NSE as we have examined in Section 2. Then it is easy to calculate its NSE and KGE: $NSE_1 = 0, KGE_1 \approx 0.5$. It is very clear that we cannot improve this simulation since the power of random noise is equal to the power of observations. But this is not true if we measure model performance by NSE or KGE. Constructing a new simulation which is half of $s_1$
$$s_2 = 0.5s_1 = 0.5o + 0.5\varepsilon, \qquad (2)$$

and calculating its NSE and KGE we obtain

$$NSE_2 = 1 - \frac{\overline{(o-s_2)^2}}{\sigma_o^2} = 1 - \frac{\overline{(0.5o-0.5\varepsilon)^2}}{\sigma_o^2} = 1 - \frac{0.5\sigma_o^2}{\sigma_o^2} = 0.5, \qquad (3)$$

$$KGE_2 = 1 - \sqrt{(\rho-1)^2 + (\sigma_2/\sigma_o - 1)^2} = 1 - \sqrt{\left(1/\sqrt{2}-1\right)^2 + \left(1/\sqrt{2}-1\right)^2} = 2 - \sqrt{2} \approx 0.6. \quad (4)$$

Suddenly, both NSE and KGE indicate that $s_2$ is better than $s_1$ considerably, although all we do is just halving $s_1$. However, Eq. (2) in nature is equivalent to Eq. (1), and we should not improve any simulation by just scaling the observations and the random error.

We have tried to find a remedy for this problem in Section 3 when we explore extended versions of NSE and KGE in the general cases. If the extension is applied, we will see that $NSE_2 = 0, KGE_2 \approx 0.5$, which show consistency with the evaluation of $s_1$. Our motivation to build the new scores is somehow unclear in the original version of the paper. Therefore, in the revised version of the paper, we intend to add this example into the revised manuscript to make our approach more rigorous.

Sincerely,
On behalf of the authors,
Le Duc.

---

## Author Comment (AC3)

**Reply to Referee #2**

We would like to thank the reviewer for the very detailed and thoughtful comments and the time that you spent on the article. Here we would like to present our point-by-point answers in blue and red colors (the red color for citations), following each of the reviewer comments reproduced in a black font.

**Major comments:**

First of all, the paper presents a link to signal processing without always explicitly mentioning i) what is known in the signal processing (or other) literature and presented here for a hydrological readership and ii) what is new. The paper demonstrates the link between correlation and what is called the here irreducible noise-to-signal ratio component for an additive error model, without clearly saying what is new about this presentation of the correlation as a measure of noisiness of the observed signal under the additive error model.

Reply 1: In Section 2 we show that the scientific meaning of NSE is truly revealed under the additive error model, which turns out to be the well-known signal-to-noise ratio in signal processing. More exactly, NSE is identical to the noise-to-signal ratio (NSR). This interpretation of NSE has been already presented in some previous studies as we have described in Introduction: "This insight was expressed clearly in Moriasi et al. (2007) when they understood NSE as the relative magnitude of variances of noise and variances of informative signals". The reviewer also pointed out that Ding (2018) has described NSE in terms of a ratio of variances: the NSE value is the ratio between the difference of residual variance and the observation variance to the observation variance. Thus, our work here is to make this similarity between two important quantities in hydrology and signal processing explicit. Furthermore, since the signal-to-noise ratio is a self-explanatory concept we do not mention any studies in the signal processing literature to explain it in detail. However, with your comment on the dependence of NSR on the strength of signals, in the revised version we will add a discussion on the dependence of NSE on the strength of signals.

The fact that NSE is identical to NSR enables us to show that the correlation coefficient $\rho$, KGE, and NSE are equivalent measures of the random NSR when model simulations have no biases. This is one of the important findings of this study since this means any measure can be used interchangeably in hydrology. In particular, we have found that $\rho$ reflects noisiness under the additive error model. Consequently, this enables us to derive a new score which we call correlation efficiency. When biases are introduced back, from $\rho$ we get back to the Nash-Ding efficiency (NDE). This connection between NDE and NSR is not easy to recognize as in the case of NSE and NSR. However, due to our uninformed assumptions in the current manuscript we do not know about the existence of NDE in the literature. In the revised version, we intend to add a description for NDE and its relationship with NSR. More importantly, the equivalence of $\rho$, KGE, and NSE in its

turn leads to another important finding: corresponding to the value NSE=0 which indicates the boundary between "bad" and "good" simulations, the values of $\rho$ and KGE are approximately 0.7 and 0.5, respectively.

As mentioned above, NSE is primarily used in hydrologic model calibration and not in forecast assessment. In classical model calibration, the simulation model is not updated with past observations of the system state and all observations are available at the time of calibration. Due to this terminology issue, some discussed points are confusing (e.g. line 95 following). Accordingly, in the present state, this paper might contribute to further confusion rather than a better understanding of NSE in other fields of environmental science where models are rarely used in a forecast setting.

Reply 2: We agree with the reviewer that NSE is mainly used in hydrology for model calibration. Our intention when writing this paper is to show that NSE as a similarity measure can be used in any scientific fields, not limited to hydrology, for the purposes of model calibration, forecast assessment, and verification. Therefore, we chose to approach NSE from the perspective of forecast assessment to show its diversity in application, even though the mathematical framework also works for other purposes. However, since this paper mainly aims to the hydrological reader, in accordance with the reviewer's comment, we will revise the manuscript content by changing its perspective from forecast assessment to model calibration. The discussion in the line 95 will be put into the context of forecast verification to avoid the possible confusion as the reviewer has pointed out.

The new proposed view in term of signal and noise leads the authors to the statement that for additive errors "when NSE goes below zero, the power of noise starts dominating the power of variation of observations"; this is what was already presented in similar terms in the work of Ding (2018) (see also his comment on the present paper in the public discussion); the NSE value is the ratio between the difference of residual variance and the observations variance to the observation variance. The relation to the "benchmark model" (being the observed mean) is nevertheless present in the hydrological model calibration literature: the requirement that a simulation should be better than the observed mean simply corresponds to the requirement that there should be more signal than noise. This is not problematic in model calibration where all observations are accessible at time of calibration (again: NSE is primarily used for calibration; in forecasting, the requirement also makes sense: the model should be better than the simplest possible forecast, which is the mean of *past* observations). Rather than **trying to show that there is no link** between the "benchmark concept" and the variance view (Ding, 2018) and the signal processing view (this paper), **it would be more interesting to show the actual link**. And it would be of prime importance to discuss the actual problem of NSE: it is not useful to compare m

Reply 3: Thank you for your suggestion. In the manuscript, we however try to link the signal processing view to the observed mean benchmark. In fact, in Section 2 we have stated that the observed mean cannot be modeled by the additive error model and we have postponed the problem

until Section 3 when we have examined the general cases. We would like to reproduce the discussion on this problem in the end of Section 3 here:

"In other words, all forecasts uncorrelated to observations, which include the mean forecast, should be classified to the worst forecast with $NSE = -\infty$. It can be justified by noticing that information on variation of observations is totally unknown if only available is an uncorrelated forecast. The generalized NSE therefore provides a new interpretation of the mean forecast. Rather than a benchmark marking the boundary between skilful and unskilful forecasts, the mean forecast is indeed the worst forecast which can be beat by any forecast correlated to observations."

Thus, the link between the benchmark concept and NSE is more sophisticated than what the traditional form of NSE apparently suggests:

$$NSE = 1 - \frac{\overline{(o-s)^2}}{\overline{(o-\mu_o)^2}}, \qquad (1)$$

where it is quite obvious that the simulation $s$ is compared against the observed mean $\mu_o$. It is no longer obvious if the extended NSE in the general cases is used instead:

$$NSE = 1 - \frac{\overline{(ao-s)^2}}{\overline{(ao-a\mu_o)^2}} = 1 - \overline{\left(\frac{o}{\sigma_o} - \frac{1}{\rho}\frac{f}{\sigma_f}\right)^2}. \quad (2)$$

Only when the multiplicative bias $a = 1$, we can identify the numerator in (2) as the mean squared error (MSE) and the corresponding denominator as the MSE of the simulation equal to the observed mean. However, the constant simulation $s = \mu_o$ cannot described by the additive error model, therefore this is not an appropriate comparison.

In order to clarify the problem, we summarize our arguments as follows:

1. In the signal processing viewpoint, our first additive error model cannot deal with the observed mean forecast.

2. In the additive error model, NSE=0 means that noise dominates informative signals, which is unrelated to the observed mean.

3. The mixed multiplicative-additive model enables us to interpret the case of the observed mean when the multiplicative bias a=0.

4. However, the traditional NSE is not robust to multiplicative biases (we shall demonstrate this later in Reply 13). When we design a new score robust to multiplicative biases, the observed mean should be interpreted as a "very bad" forecast which gives us no information about observation variability.

5. We admit that the observed mean can be easily obtained in the hydrological model calibration and seems to be reasonable as a benchmark. However, there are no meaning to choose the observed mean as a benchmark if you stand on our signal-processing viewpoint of NSE.

Of course, we can define MSE for any simulation without considering any error models between simulations and observations through the formula $MSE = \overline{(o-s)^2}$, and then NSE (1) is truly a skill score that compares MSEs of $s$ and $s = \mu_o$. However, if the additive error model is not

followed, the meaning of MSE becomes questionable with its failure in many cases as demonstrated in Wang and Bovik (2009).

Furthermore, it is important to point out (see above) that it has long been known that NSE and KGE and correlation are all equivalent in terms of identifying a model with the minimum error (smallest error variance); this is not new but identifying the solution with the smallest error variance is rarely the single main objective of hydrologic model calibration (read e.g. key model estimation literature in hydrology by Keith Beven or Hoshin Gupta).

Reply 4: In our understanding, we have not yet seen any studies showing that "NSE and KGE and correlation are all equivalent in terms of identifying a model with the minimum error". For example, we would like to cite here some recent studies:

Knoben et al., 2019: "We first show this in mathematical terms and then present results from a synthetic experiment to highlight that NSE and KGE values are not directly comparable and that understanding of the NSE metric does not translate well into understanding of the KGE metric".

Lamontagne et al., 2020: "In general, the two theoretical statistics $E$ and $E'$ are only very roughly equal when $\alpha = \rho$, which would be the case for simple linear regression or when the simulations are independent of the model residuals. Otherwise, for more realistic and complicated models with $\alpha \neq \rho$, the values of $E$ and $E'$ can be expected to differ and quite significantly so".

(Here we would like to add some explanations for $E$ and $E'$. These two quantities are two probabilistic statistics of NSE and KGE corresponding to their more familiar estimators used in the literature).

Clark et al., 2021: "KGE is only loosely related to NSE and thus MSE, with a complex relationship between NSE and KGE that depends on several factors. For general cases, the relationship between NSE and KGE depends on the coefficient of variation of the observations… In the special case of unbiased models, the relationship between NSE and KGE still remains complex".

Since the finding that $\rho$, KGE, and NSE are equivalent measures of NSR when model simulations have no biases has been considered one of the important findings in our study. If this finding has been known long before, we need to rewrite the manuscript and cite the original papers that pointed out this fact. Therefore, we are very grateful if the reviewer can inform us such important papers.

The paper omits to discuss one of the key issues with using NSE for model performance assessment: if the signal is very strong, it is easy to obtain a good signal to noise ratio – i.e. easy to achieve good model performance; in other words, NSE should not be used to compare different case studies (different modelled signals), a problem that is still not sufficiently recognized when using NSE. This problem leads to many erroneous conclusions of the type: "NSE is better for simulation of signal A than for simulation of signal B – we can conclude that the model can better reproduce signal A than signal B".

Reply 5: Thank you for pointing out this important point. As we have stated in Reply 1, we will add a discussion on this problem in the revised version.

The part that is interesting from my viewpoint is the part on a score that is invariant under any general translation; from a hydrological process modelling view point, however, we would need to be able to assign the translation to a clear physical phenomenon such as measurement bias. The definition of an upper limit for NSE under translation (i.e. bias correction) and the explicit link between NSE>= 0 and a corresponding correlation value (rho>=0.7) (under an additive error model) is certainly also interesting. This could open new ways of comparing model performance for different case studies for example.

Reply 6: Thank you for your suggestion. Since we have shown that $\rho$, KGE, and NSE are equivalent measures of NSR when model simulations have no biases, and NSR tends to be small for strong signals and large for weak signals (all three measures tend to favor strong signals), we think that it is still impossible to use $\rho$ to compare model performance for different case studies. Here the same argument is applied as in your previous comment.

However, I do not think it is appropriate to assign this new skill score a name related to NSE – an abbreviation that should be exclusively used for the original formulation. Any new skill score should have a new name.

Reply 7: Thank you for your suggestion. We have shown that any monotonic function of $\rho$ can generate an equivalent measure of the random NSR. Since there are many such functions, it is impossible to name all resulting scores. Therefore, we only focus on three special scores that lead back to NSE, KGE, and NDE (or $\rho$). Due to this relationship to the three common scores in hydrology, we still use these traditional names in the text.

I think the term forecast should not be used in this paper; in hydrology, today, forecast does not have the general meaning of "a simulated value" but is related to real-time forecasting; in general, there is an unclear use of the term forecast; most hydrological models are not used in a forecasting mode (predicting future states based on the current observed state) but in a simulation mode starting with initial conditions and observed inputs only; accordingly, the discussion related to forecast availability in lines 56 is misleading; NSE is primarily used in model calibration and not in forecast skill assessment; in calibration, the availability of the observations is a pre-condition.

Reply 8: As we have stated in Reply 2, we will revise the manuscript in the perspective of model calibration.

Similarly, I would avoid "skillful forecast"; in hydrology, NSE is used in the context of model calibration and performance assessment but not primarily to judge the skill of forecasts.

Reply 9: As we have stated in Reply 2, we will revise the manuscript in the perspective of model calibration and discard the use of "skillful forecast".

What is meant by "and this has no relation with the benchmark forecast that is equal to the mean of

observations"; obviously the value of NSE=0 has a very clear relation to the benchmark since if the all simulations equal the mean of the observations, we get NSE=0.

Reply 10: We have explained this sophisticated problem in Reply 3. Here the point is that we implicitly change the signal processing viewpoint to the skill score viewpoint when we interpret the case NSE=0. Whereas the observed mean is not allowed under the former, it is a trivial forecast under the latter. However, if the skill score viewpoint assumes the additive error model so that MSE makes sense, the observed mean has to be excluded as in the former case.

Another argument for not using the observed mean as the benchmark forecast is that any constant forecast such as the observed mean is the weakest signal (no variation at all). However, it is well-known that NSE should not be applied for weak signals, e.g., low-flow discharges, because NSR can be very large for weak signals.

What is meant by "Corresponding to NSE=0, the critical values of KGE is given approximately by 0.5."; how do you define the critical value?

Reply 11: By the critical value of KGE, we mean the value of KGE indicates the boundary between "good" and "bad" simulation corresponding to the value zero in case of NSE. We will rewrite this sentence to avoid confusion in the revised version.

"In the general cases, when the additive error model is replaced by a mixed adaptive multiplicative error model, the traditional NSE is shown not to be a well-defined notion": The notion of NSE cannot depend on the error model since NSE characterizes the model performance independent of an error model assumption; how could e.g. the notion of a bias depend on the error model assumption?

Reply 12: By saying that "the traditional NSE is shown not to be a well-defined notion" we mean NSE when conceived as NSR is not a well-defined notion. Our rationale in our study is that NSE only makes sense under the additive error model, and its meaning truly depends on the underlying error model. The situation is very similar to the case of MSE as examined in Wang and Bovik (2009). Outside the scope of the additive error model, we will see the failure of NSE as a measure for model performance, or even the contradiction when using NSE as we shall show in the following reply. Similarly, for the notion of bias, we can easily show that the notion of bias truly depends on the error model assumption. Let us consider the following error model for rainfall simulations:

$$s = \varepsilon * o, \qquad (3)$$

where $\varepsilon$ denotes random multiplicative bias. Then if we define $\overline{s - o}$ as the bias, this notion will make no sense at all under the error model (3).

Why do we need the requirement that "the generalized NSE is invariant under affine transformations"? This is your requirement, not the hydrologist's requirement.

Reply 13: This requirement is established under the premise that we should not have any contradiction in model evaluation. Our approach is to impose important invariances on measures

like NSE and KGE so that they work in any situation enabled by the error model without resulting in contradiction. This is our philosophy and it might be possible that some hydrologists may think that contradiction if any is practically acceptable (an example shall be shown below). However, we believe that many hydrologists want to use scores invariant under affine transformations if they assume the mixed additive-multiplicative error model since this requirement is necessary to avoid any contradictions in their model evaluation. Naturally, this requirement is the consequence of the mixed additive-multiplicative error model:

$$s = ao + b + \varepsilon. \quad (4)$$

When the underlying error model is additive ($a = 1$ in (4)), the only requirement is translation-invariant.

We now illustrate that if NSE is not invariant under affine transformations, we will get contradiction. This also answer your previous comment on the failure of NSE when it is used outside of the additive error model.

Let us consider a model simulation with an additive random error

$$s_1 = o + \varepsilon, \quad (5)$$

where we assume $\mu_{s1} = \mu_o = 0$, and $\sigma_o = \sigma_e$. This simulation in deed gives us the boundary between "good" and "bad" simulations from the viewpoint of NSE as we have examined in Section 2. Then it is easy to calculate its NSE and KGE: $NSE_1 = 0, KGE_1 \approx 0.5$. It is very clear that we cannot improve this simulation since the power of random noise is equal to the power of observations. But this is not true if we measure model performance by NSE or KGE. Constructing a new simulation which is half of $s_1$

$$s_2 = 0.5s_1 = 0.5o + 0.5\varepsilon, \quad (6)$$

and calculating its NSE and KGE we obtain

$$NSE_2 = 1 - \frac{\overline{(o-s_2)^2}}{\sigma_o^2} = 1 - \frac{\overline{(0.5o-0.5\varepsilon)^2}}{\sigma_o^2} = 1 - \frac{0.5\sigma_o^2}{\sigma_o^2} = 0.5, \quad (7)$$

$$KGE_2 = 1 - \sqrt{(\rho - 1)^2 + (\sigma_2/\sigma_o - 1)^2} = 1 - \sqrt{\left(1/\sqrt{2} - 1\right)^2 + \left(1/\sqrt{2} - 1\right)^2} = 2 - \sqrt{2} \approx 0.6. \quad (8)$$

Suddenly, both NSE and KGE indicate that $s_2$ is better than $s_1$ considerably, although all we do is just halving $s_1$. However, Eq. (6) in nature is equivalent to Eq. (5), and we should not improve any simulation by just scaling the observations and the random error.

Conclusion: "Its choice is dictated by the fact that at this value the power of noise starts dominating the power of variation of observations." Who dictates it? Why would this interpretation be superior to previous explanations?

Reply 14: We will modify this sentence in the revised version. And for this new interpretation in deed forms a sophisticated problem that we have explained in the third reply.

Why would we need to adjust NSE for other error models? It was never intended to be used in

conjunction with error models but to yield an easy to interpret performance measure; most authors do not specify an error model; this difference should become clear.

Reply 15: We have explained this sophisticated problem in the Replies 12, 13, 14. In short, we cannot have a universal measure for all cases. The traditional NSE makes sense under the additive error model. However, it is easy to show its failure when other error models are assumed as demonstrated in our example in Reply 13. The situation here is very similar to the case of MSE as examined in Wang and Bovik (2009). Since most authors do not specify an error model explicitly as the reviewer said, we have tried to secure this problem by finding more general versions of NSE and KGE that are applicable in most cases. These efficiencies are described in Section 3 of the manuscript.

Regarding the apparent debate on the meaning of Nash: this needs a reference; did someone else say this or is this your interpretation? I have never heard / seen anyone saying that there is a discussion about what Nash means; there is simply no need to use Nash in other disciplines.

Reply 16: Here we mean that there existed different interpretations for the scientific meaning of NSE. Later in Introduction we have listed three distinct interpretations including reference lists: (1) NSE as the coefficient of determination $R^2$ in linear regression, (2) NSE as a skill score for MSE, and (3) NSE as a multi-criteria score.

**Minor comments:**

Line 9-10: I do not have the original terminology of Nash and Sutcliffe in my head but: NSE is mostly used in model calibration not in forecast quality assessment; in hydrology, a forecast is the prediction of a future state as a function of observed past states; this is a special use of a simulation model. This paper needs reformulating the text in terms of simulations models (input-output models) rather than in terms of forecasts.

Reply: As we have stated in Reply 2, we will revise the manuscript in the perspective of model calibration.

Line 23-24: bad use of parentheses

Reply: Thank you for pointing out all typographical and grammatical errors. We will apply proofread for the revised version.

Line 24: what is the benchmark? this is not a term with a universal meaning; what benchmark are you talking about?

Reply: Thank you. We will rewrite this sentence.

Line 29: as far as I see, the main point of that paper was about the fact that NSE values of different case studies should not be compared; it would have been interesting to reflect on this problem in the present paper.

Reply: As we have stated in Reply 1, we will add a discussion on this problem in the revised version.

Line 33-34: this needs a reference; did someone else say this or is this your interpretation? I have never heard / seen anyone saying that there is a discussion about what Nash means;

Reply: Again, here we mean that there existed different interpretations for the scientific meaning of NSE. Later in Introduction we have listed three distinct interpretations including reference lists: (1) NSE as the coefficient of determination $R^2$ in linear regression, (2) NSE as a skill score for MSE, and (3) NSE as a multi-criteria score.

Line 68: strange to give a reference posterior to the KGE suggestion

Reply: We will come back to the original KGE in the revised version.

Line 74-75: what is meant, please revise the sentence

Reply: Thank you. We will rewrite this sentence.

Line 88-89: what does this mean? it is not the error model that excludes forecasts and what is "including the mean forecast"? what is probably meant: With an additive error model, the observational variance is decomposed into the variance of the simulation model plus the variance of the error model; accordingly, the simulation model variance has to be lower than the observational

Reply: Thank you. We will rewrite this sentence with your suggestion.

Line 112: how do you obtain expectation of o^2 = mean^2 + variance; it is probably obvious to most readers, but why not still stay that it follows from the definition of the variance?

Reply: Thank you. We will modify the formula.

Line 157: forecaster are not all men, this is not appropriate in modern scientific writing

Reply: Thank you. We will rewrite this sentence.

Line 187: more than what?

Reply: This means that expression of NSE in terms of $\rho$ better than in terms of RNSR. We will rewrite this sentence.

Line 207: wrong grammar, unclear what the sentence wants to say;

Line 255-256: misleading formulation, here and later: the error model does not exert some sort of selection, it does not exclude; I suggest: "the additive error model implies that the simulations have

to a smaller variance than the observations since part of the observation variation is assigned to the error, not to the simulation". besides: this is of course a well-known fact in the hydrological literature dealing with Bayesian model inference (where the model error is inferred along with the model parameters, e.g. paper by Dmitri Kavetski

Reply: Thank you. We will rewrite this sentence with your suggestion.

Line 416: s

Line 420: adding values, what does this mean, adding new samples to the time series? Unclear

Reply: This means that the values of NSE can be easily changed under translations of simulations and observations. We will rewrite this sentence.

Line 422: of

Line 423: ls

Line 428: no, where did you see this? I do not remember having seen this before

Reply: We have cited some researches in Section 2 on this problem:

This relatively large gap can lead to misjudgement on forecast performances in practice since similar to NSE, modelers tend to consider KGE=0 as the boundary value between skilful and unskilful forecasts (Anderson et al., 2017; Fowler et al., 2018; Siqueira et al., 2018; Sutanudjaja et al., 2018; Towner et al., 2019). Thus, all forecasts with KGE between [0,0.5] are wrongly classified to be good forecasts while they are indeed unskilful forecasts. It is worth noting that Rogelis et al. (2016) assigned the value KGE=0.5 to be the threshold below which forecasts are considered to be "poor".

Line 431-432: what does this sentence mean?

Reply: Thank you. We will rewrite this sentence with your above suggestion.

Line 432-433: what is "a distribution between forecasts and observations", what does this mean?

Reply: We mean the joint probability distribution between forecasts and observations.

Line 436-437: what is "it"; what means "it is found to be the traditional"?

Reply: "it" is referred to the generalized NSE.

Line 438: bad writing practice: this use of parentheses is a mis-use of parentheses

Reference

Clark, M. P., Vogel, R. M., Lamontagne, J. R., Mizukami, N., Knoben, W. J. M., Tang, G., et al.: The abuse of popular performance metrics in hydrologic modeling. *Water Resources Research*, *57*, e2020WR029001, doi:10.1029/2020WR029001, 2021.

Knoben, W. J. M., Freer, J. E., and Woods, R. A.: Technical note: Inherent benchmark or not? Comparing Nash–Sutcliffe and Kling–Gupta efficiency scores. *Hydrol. Earth Syst. Sci.*, **23**, 4323–4331, doi:10.5194/hess-23-4323-2019, 2019.

Lamontagne, J. R., Barber, C. A., and Vogel, R. M.: Improved estimators of model performance efficiency for skewed hydrologic data. *Water Resources Res.*, **56**, e2020WR027101, doi:10.1029/2020WR027101, 2020.

Wang, Z., and Bovik, A. C.: Mean squared error: Love it or leave it? A new look at signal fidelity measures. *IEEE Signal Processing Magazine*, **26**(1), 98–117, doi:10.1109/msp.2008.930649, 2009.

Sincerely,
On behalf of the authors,
Le Duc.

---

## Author Response (AR1)

**Response to reviewers**

We would like to thank the two reviewers and Dr. Ding for the detailed and thoughtful comments and the time that you spent on the article. Here we would like to present a common answer for the reviewers. We respond to each specific question by each reviewer later. We would like to present our point-by-point answer in blue and red color, following each of the reviewer comments reproduced in black font. With our revision we hope that the reviewers will find it a better article.

Therefore, we have modified the manuscript as follows:

- Since this paper mainly aims to the hydrological reader, in accordance with the reviewer's comment, we have revised the manuscript content by changing its perspective from forecast assessment to model calibration. Also, all mathematical formulas that use the symbol f (to represent forecasts) have been updates to use the symbol s (to represent simulations).

- The introduction: the Nash-Ding efficiency (NDE) has been added into the introduction to show its closed relationship with the Nash-Sutcliffe efficiency (NSE). Since NDE is a dual of NSE from the regression viewpoint, we will use the two scores interchangeably in the paper to examine their scientific meanings.

- Section 2 "Additive error models": We have added two important implications of the signal processing viewpoint on NSE: (1) the dependence of NSE on the strength of signals implies NSE should not be used to compare different signals; (2) the signal-to-noise ratio suggests a natural way to define a threshold for good/bad model distinction, which does not require any benchmark model as in the traditional interpretation of the threshold NSE=0. The relationship between NSE and the Kling-Gupta efficiency now covers also NDE.

- Section 3 "Mixed multiplicative-additive error models": An important example on the failure of the traditional efficiencies when applied outside of the additive error model has been added to make our motivation for the development of new efficiencies clearer. The link between the benchmark simulation that is equal to the observed mean and the signal processing viewpoint on different efficiencies has been shown to be a sophisticated problem. To clarify this problem, the concise arguments underlying this link has been added into the end of this section.

- The conclusion: this section has been updated to reflect all the changes of the manuscript. Especially, the findings in the end of this section have been rewritten to make them more accessible.

Now, we would like to reply in detail each question of the reviewers.

Reviewer 1:

The paper deals with a novel interpretation of the NSE score measure starting from the observation that it is mostly used in hydrology and poorly exploited in other sciences.

This interpretation is based on a signal processing viewpoint.

While the paper is interesting and useful the following concerns are at the hand.

The NSE interpretation provided is based on a model error for forecast which is given in eq 5 and is basically driven by a Gaussian random error being it equal to noise in the signal processing viewpoint. This basic foundation provides by itself a lack of generality with respect to application of NSE to other sciences included hydrology. In hydrology the NSE is intended as a model performance metric where the difference between model and observation is not limited to noise. Differences between model and observation but also differences between observation and reality and also differences between different models can be analyzed by means of NSE or KGE whose meaning is quite clear and leaves no doubts in my personal opinion.

In a hydrologic model, but also in other earth sciences, different models may arise because different processes are modelled in a stochastic or determinist way and/or because some processes are described or neglected following the fact that they can be more or less important according to the time-space scale of application and modeling purpose. Hence the difference between model output and observations may be very different from what given in eq 5, It can be deterministic or stochastic, and affected by deterministic or stochastic (or both) variability.

As a consequence, it seems that the proposed analysis, while interesting and well founded in the context of the signal processing field (or any other fields where only noise provides difference between model output and observation). In the same light one may not accept the "general case" version of NSE which is obtained by considering the multiplicative error, beside the additive error, defined in eqs (32). Even in this case the "general case" should be addressed as relative to which field of application, besides the field of signal processes or affine methods.

I believe the authors should strongly address this issue in a revised version of the manuscript.

Reply: First of all, we totally agree with the reviewer on the point that the additive error model that we consider in Section 2 is relatively limited and cannot cover all complicated relationships between models and observations in reality. Our purpose when using this additive error model is to show that the scientific meaning of NSE is truly revealed under this error model, which turns out to be the well-known signal-to-noise ratio in signal processing.

Due to the limit of the additive error model, in Section 3 we consider more general cases which can be described by joint probability distributions of model simulations and observations. Although the reviewer mentioned that general cases have not been dealt, we believe that our mixed multiplicative-additive error model addressed a wide range of cases of model-observation fitting. We would like to emphasize here that any relationship between model simulations and observations

can always be analyzed through their joint probability distributions. Then we assume that such distributions are bi-variate normal distributions, which is usually observed in reality. Again, we would like to emphasize that we do not assume the mixed multiplicative-additive error model for the general cases. Rather than that, this multiplicative-additive error model is a consequence of the bi-variate normal distribution of model simulations and observations.

Of course, if this joint distribution is not Gaussian, we cannot expect to see the mixed multiplicative-additive error model and the relationship between model simulations and observations may be even more complicated as the reviewer commented. However, we believe that there are some suitable transformations to make observations and simulations follow the Gaussian distribution in most cases including hydrological applications. The generality of our error model described above has already been discussed in the manuscript:

"Since all information on simulations and observations is encapsulated in their joint probability distribution, we can seek the general form of this conditional distribution from their joint distribution in the general cases. For this purpose, we will assume that this joint probability distribution is a bivariate normal distribution. If the joint distribution is not Gaussian, we need to apply some suitable transformations to $s, o$ such as the root squared transformation $(s, o) \rightarrow (\sqrt{s}, \sqrt{o})$, the log transformation $(s, o) \rightarrow (\log(s), \log(o))$, the inverse transformation $(s, o) \rightarrow (1/s, 1/o)$, … (Pushpalatha et al., 2012)."

We have shared the same opinion with the reviewer that the meanings of NSE and KGE "is quite clear and leaves no doubts" until we have come up with a counterexample based on the mixed multiplicative-additive error model. In the revised version of the paper, we have added this example into the revised manuscript to make our motivation clearer:

"Let us consider a model simulation with an additive random error

$$s_1 = o + \varepsilon, \qquad (36)$$

where we assume $\mu_{s1} = \mu_o = 0$, and $\sigma_o = \sigma_e$. This simulation in deed gives us the thresholds $NSE_1 = 0, NDE_1 = 0.5, KGE_1 = 0.5$ that distinguish good simulations from bad ones as we have examined in Sect. 2. It is very clear that we cannot improve this simulation since the power of random noise is equal to the power of observations. But this is not true if we measure performances with NSE, NDE, and KGE by constructing a new simulation which is half of $s_1$

$$s_2 = 0.5s_1 = 0.5o + 0.5\varepsilon. \qquad (37)$$

Calculating its NSE and KGE, we obtain

$$NSE_2 = 1 - \frac{\overline{(o - s_2)^2}}{\sigma_o^2} = 1 - \frac{\overline{(0.5o - 0.5\varepsilon)^2}}{\sigma_o^2} = 1 - \frac{0.5\sigma_o^2}{\sigma_o^2} = 0.5, \qquad (38a)$$

$$NDE_2 = 1 - \frac{\overline{(s_2 - o)^2}}{s_2^2} = 1 - \frac{\overline{(0.5\varepsilon - 0.5o)^2}}{(0.5o + 0.5\varepsilon)^2} = 1 - \frac{0.5\sigma_o^2}{0.5\sigma_o^2} = 0., \qquad (38b)$$

$$KGE_2 = 1 - \sqrt{(\rho - 1)^2 + (\sigma_2/\sigma_o - 1)^2} = 1 - \sqrt{\left(1/\sqrt{2} - 1\right)^2 + \left(1/\sqrt{2} - 1\right)^2} = 2 - \sqrt{2} \approx 0.6 \quad .$$

$$(38c)$$

Suddenly, NSE and KGE indicate that $s_2$ is better than $s_1$ considerably, although all we do is just halving $s_1$. In contrast, NDE gives a very different evaluation: $s_2$ is much worse than $s_1$. However, (37) in nature is equivalent to (36), and we should not make any simulation better or worse by just scaling the observations and the random error. This simple example is enough to show that the scientific meaning of the traditional scores like NSE becomes questionable when we introduce multiplicative biases into the error model."

We have tried to find a remedy for this problem in Section 3 when we explore extended versions of NSE and KGE in the general cases. If the extension is applied, we will see that $NSE_2 = 0, NDE_2 = 0, KGE_2 \approx 0.5$, which show consistency with the evaluation of $s_1$.

Reviewer 2:

**Major comments:**

First of all, the paper presents a link to signal processing without always explicitly mentioning i) what is known in the signal processing (or other) literature and presented here for a hydrological readership and ii) what is new. The paper demonstrates the link between correlation and what is called the here irreducible noise-to-signal ratio component for an additive error model, without clearly saying what is new about this presentation of the correlation as a measure of noisiness of the observed signal under the additive error model.

Reply 1: In Section 2 we show that the scientific meaning of NSE is truly revealed under the additive error model, which turns out to be the well-known signal-to-noise ratio in signal processing. More exactly, NSE is identical to the noise-to-signal ratio (NSR). This interpretation of NSE has been already presented in some previous studies as we have described in Introduction: "This insight was expressed clearly in Moriasi et al. (2007) when they understood NSE as the relative magnitude of variances of noise and variances of informative signals". The reviewer also pointed out that Ding (2018) has described NSE in terms of a ratio of variances: the NSE value is the ratio between the difference of residual variance and the observation variance to the observation variance. Thus, our work here is to make this similarity between two important quantities in hydrology and signal processing explicit. Furthermore, since the signal-to-noise ratio is a self-explanatory notion, we do not mention any studies in the signal processing literature to explain it in detail. However, as we have stated in the common answer, we have added a discussion on the dependence of NSE on the strength of signals in the revised version, and a discussion on a natural threshold for good/bad model distinction from the viewpoint of the signal-to-noise ratio.

The fact that NSE is identical to NSR enables us to show that the correlation coefficient $\rho$, KGE, NDE, and NSE are equivalent measures of the random NSR when model simulations have no biases. This is one of the important findings of this study since this means any measure can be used

interchangeably in hydrology. Consequently, this enables us to derive any score only using $\rho$, and this is described into Section 2.3.

As mentioned above, NSE is primarily used in hydrologic model calibration and not in forecast assessment. In classical model calibration, the simulation model is not updated with past observations of the system state and all observations are available at the time of calibration. Due to this terminology issue, some discussed points are confusing (e.g. line 95 following). Accordingly, in the present state, this paper might contribute to further confusion rather than a better understanding of NSE in other fields of environmental science where models are rarely used in a forecast setting.

Reply 2: We agree with the reviewer that NSE is mainly used in hydrology for model calibration. Our intention when writing this paper is to show that NSE as a similarity measure can be used in any scientific fields, not limited to hydrology, for the purposes of model calibration, forecast assessment, and verification. Therefore, we chose to approach NSE from the perspective of forecast assessment to show its diversity in application, even though the mathematical framework also works for other purposes. As we have stated in the common answer, we have revised the manuscript in the perspective of model calibration. The discussion in the line 95 will be put into the context of forecast verification to avoid the possible confusion as the reviewer has pointed out.

The new proposed view in term of signal and noise leads the authors to the statement that for additive errors "when NSE goes below zero, the power of noise starts dominating the power of variation of observations"; this is what was already presented in similar terms in the work of Ding (2018) (see also his comment on the present paper in the public discussion); the NSE value is the ratio between the difference of residual variance and the observations variance to the observation variance. The relation to the "benchmark model" (being the observed mean) is nevertheless present in the hydrological model calibration literature: the requirement that a simulation should be better than the observed mean simply corresponds to the requirement that there should be more signal than noise. This is not problematic in model calibration where all observations are accessible at time of calibration (again: NSE is primarily used for calibration; in forecasting, the requirement also makes sense: the model should be better than the simplest possible forecast, which is the mean of *past* observations). Rather than trying to show that there is no link between the "benchmark concept" and the variance view (Ding, 2018) and the signal processing view (this paper), it would be more interesting to show the actual link. And it would be of prime importance to discuss the actual problem of NSE: it is not useful to compare m

Reply 3: Thank you for your suggestion. In the manuscript, we however try to link the signal processing view to the observed mean benchmark. In fact, in Section 2 we have stated that the observed mean cannot be modeled by the additive error model and we have postponed the problem until Section 3 when we have examined the general cases. We would like to reproduce the discussion on this problem in the end of Section 3 here:

"In other words, all simulations uncorrelated to observations, which include the observed mean, should be classified to the worst simulations with $NSE = -\infty$. It can be justified by noticing that

information on variation of observations is totally unknown if only available is an uncorrelated simulation. The generalized NSE therefore provides a new interpretation of the benchmark simulation $s = \mu_o$. Rather than a benchmark marking the boundary between good and bad simulations, the observed mean is indeed the worst simulation which can be beat by any simulations correlated to observations."

Then under the mixed multiplicative-additive error model, the link between the benchmark concept and NSE is more sophisticated than what the traditional form of NSE apparently suggests:

$$NSE = 1 - \frac{\overline{(o-s)^2}}{\overline{(o-\mu_o)^2}}, \qquad (1)$$

where it is quite obvious that the simulation $s$ is compared against the observed mean $\mu_o$. It is no longer obvious if the generalized NSE in the general cases is used instead:

$$NSE = 1 - \frac{\overline{(ao-s)^2}}{\overline{(ao-a\mu_o)^2}} = 1 - \overline{\left(\frac{o}{\sigma_o} - \frac{1}{\rho}\frac{f}{\sigma_f}\right)^2}. \quad (2)$$

Only when the multiplicative bias $a = 1$, we can identify the numerator in (2) as the mean squared error (MSE) and the corresponding denominator as the MSE of the simulation equal to the observed mean. However, the constant simulation $s = \mu_o$ cannot described by the additive error model, therefore this is not an appropriate comparison.

In order to clarify the problem, we summarize our arguments as follows in the revised manuscript:

- Under the perspective of signal processing, the additive error model cannot deal with the benchmark model $s = \mu_o$.
- In the additive error model, NSE=0 means that noise dominates informative signals, which is unrelated to the observed mean.
- The mixed multiplicative-additive model enables us to interpret the case of the observed mean when the multiplicative bias a=0.
- However, the traditional NSE is not robust to multiplicative biases. When we design a new score robust to multiplicative biases, the observed mean should be interpreted as the worst simulation which gives us no information on observation variability.
- Although the observed mean can be easily obtained in hydrological model calibration and seems to be reasonable as a benchmark, it makes no sense to choose the observed mean as a benchmark simulation from the signal-processing viewpoint of NSE.

Of course, we can define MSE for any simulation without considering any error models between simulations and observations through the formula $MSE = \overline{(o-s)^2}$, and then NSE is truly a skill score that compares MSEs of $s$ and the constant simulation $s = \mu_o$. However, if the additive error model is not followed, the meaning of MSE becomes questionable with its failure in many cases as demonstrated in Wang and Bovik (2009). Furthermore, the use of the observed mean benchmark will entail $NDE = -\infty$, which is clearly a failure of this approach for NDE.

Furthermore, it is important to point out (see above) that it has long been known that NSE and KGE

and correlation are all equivalent in terms of identifying a model with the minimum error (smallest error variance); this is not new but identifying the solution with the smallest error variance is rarely the single main objective of hydrologic model calibration (read e.g. key model estimation literature in hydrology by Keith Beven or Hoshin Gupta).

Reply 4: In our understanding, we have not yet seen any studies showing that "NSE and KGE and correlation are all equivalent in terms of identifying a model with the minimum error". For example, we would like to cite here some recent studies:

Knoben et al., 2019: "We first show this in mathematical terms and then present results from a synthetic experiment to highlight that NSE and KGE values are not directly comparable and that understanding of the NSE metric does not translate well into understanding of the KGE metric".

Lamontagne et al., 2020: "In general, the two theoretical statistics $E$ and $E'$ are only very roughly equal when $\alpha = \rho$, which would be the case for simple linear regression or when the simulations are independent of the model residuals. Otherwise, for more realistic and complicated models with $\alpha \neq \rho$, the values of $E$ and $E'$ can be expected to differ and quite significantly so".

(Here we would like to add some explanations for $E$ and $E'$. These two quantities are two probabilistic statistics of NSE and KGE corresponding to their more familiar estimators used in the literature).

Clark et al., 2021: "KGE is only loosely related to NSE and thus MSE, with a complex relationship between NSE and KGE that depends on several factors. For general cases, the relationship between NSE and KGE depends on the coefficient of variation of the observations... In the special case of unbiased models, the relationship between NSE and KGE still remains complex".

Since the finding that $\rho$, KGE, NDE, and NSE are equivalent measures of NSR when model simulations have no biases has been considered one of the important findings in our study. If this finding has been known long before, we need to rewrite the manuscript and cite the original papers that pointed out this fact. Therefore, we are very grateful if the reviewer can inform us such important papers.

The paper omits to discuss one of the key issues with using NSE for model performance assessment: if the signal is very strong, it is easy to obtain a good signal to noise ratio – i.e. easy to achieve good model performance; in other words, NSE should not be used to compare different case studies (different modelled signals), a problem that is still not sufficiently recognized when using NSE. This problem leads to many erroneous conclusions of the type: "NSE is better for simulation of signal A than for simulation of signal B – we can conclude that the model can better reproduce signal A than signal B".

Reply 5: Thank you for pointing out this important point. As we have stated in the common answer, we have added a discussion on this problem in the revised version.

The part that is interesting from my viewpoint is the part on a score that is invariant under any general translation; from a hydrological process modelling view point, however, we would need to be able to assign the translation to a clear physical phenomenon such as measurement bias. The

definition of an upper limit for NSE under translation (i.e. bias correction) and the explicit link between NSE>= 0 and a corresponding correlation value (rho>=0.7) (under an additive error model) is certainly also interesting. This could open new ways of comparing model performance for different case studies for example.

Reply 6: Thank you for your suggestion. Since we have shown that $\rho$, KGE, NDE, and NSE are equivalent measures of NSR when model simulations have no biases, and NSR tends to be small for strong signals and large for weak signals (all four measures tend to favor strong signals), we think that it is still impossible to use $\rho$ to compare model performance for different case studies. Here the same argument is applied as in your previous comment.

However, I do not think it is appropriate to assign this new skill score a name related to NSE – an abbreviation that should be exclusively used for the original formulation. Any new skill score should have a new name.

Reply 7: Thank you for your suggestion. We have shown that any monotonic function of $\rho$ can generate an equivalent measure of the random NSR. Since there are many such functions, it is impossible to name all resulting scores. Therefore, we only focus on three special scores that lead back to NSE, KGE, and NDE (or $\rho$). Due to this relationship to the three common scores in hydrology, we still use these traditional names in the text.

I think the term forecast should not be used in this paper; in hydrology, today, forecast does not have the general meaning of "a simulated value" but is related to real-time forecasting; in general, there is an unclear use of the term forecast; most hydrological models are not used in a forecasting mode (predicting future states based on the current observed state) but in a simulation mode starting with initial conditions and observed inputs only; accordingly, the discussion related to forecast availability in lines 56 is misleading; NSE is primarily used in model calibration and not in forecast skill assessment; in calibration, the availability of the observations is a pre-condition.

Reply 8: As we have stated in the common answer, we have revised the manuscript in the perspective of model calibration.

Similarly, I would avoid "skillful forecast"; in hydrology, NSE is used in the context of model calibration and performance assessment but not primarily to judge the skill of forecasts.

Reply 9: As we have stated in the common answer, we have revised the manuscript in the perspective of model calibration and discard the use of "skillful forecast".

What is meant by "and this has no relation with the benchmark forecast that is equal to the mean of observations"; obviously the value of NSE=0 has a very clear relation to the benchmark since if the all simulations equal the mean of the observations, we get NSE=0.

Reply 10: We have explained this sophisticated problem in Reply 3. Here the point is that we implicitly change the signal processing viewpoint to the skill score viewpoint when we interpret the case NSE=0. Whereas the observed mean is not allowed under the former, it is a trivial simulation

under the latter. However, if the skill score viewpoint assumes the additive error model so that MSE makes sense, the observed mean has to be excluded as in the former case.

Another argument for not using the observed mean as the benchmark simulation is that any constant simulation such as the observed mean is the weakest signal (no variation at all). However, it is well-known that NSE should not be applied for weak signals, e.g., low-flow discharges, because NSR can be very large for weak signals.

What is meant by "Corresponding to NSE=0, the critical values of KGE is given approximately by 0.5."; how do you define the critical value?

Reply 11: By the critical value of KGE, we mean the value of KGE indicates the threshold for good/bad model distinction corresponding to the value zero in case of NSE. We have rewritten this sentence to avoid confusion in the revised version:

"The scientific meaning of NSE suggests a natural way to define NSE=0 as the threshold for good/bad model distinction, and this has no relation with the benchmark simulation that is equal to the observed mean. Corresponding to NSE=0, the threshold of KGE is given approximately by 0.5."

"In the general cases, when the additive error model is replaced by a mixed adaptive multiplicative error model, the traditional NSE is shown not to be a well-defined notion": The notion of NSE cannot depend on the error model since NSE characterizes the model performance independent of an error model assumption; how could e.g. the notion of a bias depend on the error model assumption?

Reply 12: By saying that "the traditional NSE is shown not to be a well-defined notion" we mean NSE when conceived as NSR is not a well-defined notion. Our rationale in our study is that NSE only makes sense under the additive error model, and its meaning truly depends on the underlying error model. The situation is very similar to the case of MSE as examined in Wang and Bovik (2009). Outside the scope of the additive error model, we will see the failure of NSE as a measure for model performance, or even the contradiction when using NSE.

Similarly, for the notion of bias, we can easily show that the notion of bias truly depends on the error model assumption. Let us consider the following error model for rainfall simulations:

$$s = \varepsilon * o, \qquad (3)$$

where $\varepsilon$ denotes random multiplicative bias. Then if we define $\overline{s-o}$ as the bias, this notion will make no sense at all under the error model (3).

Why do we need the requirement that "the generalized NSE is invariant under affine transformations"? This is your requirement, not the hydrologist's requirement.

Reply 13: This requirement is established under the premise that we should not have any contradiction in model evaluation. Our approach is to impose important invariances on measures like NSE and KGE so that they work in any situation enabled by the error model without resulting in contradiction. This is our philosophy and it might be possible that some hydrologists may think

that contradiction if any is practically acceptable (an example shall be shown below). However, we believe that many hydrologists want to use scores invariant under affine transformations if they assume the mixed additive-multiplicative error model since this requirement is necessary to avoid any contradictions in their model evaluation. Naturally, this requirement is the consequence of the mixed additive-multiplicative error model:

$s = ao + b + \varepsilon.$ (4)

When the underlying error model is additive ($a = 1$ in (4)), the only requirement is translation-invariant.

We now illustrate that if NSE is not invariant under affine transformations, we will get contradiction. This also answers your previous comment on the failure of NSE when it is used outside of the additive error model. This example has been reproduced from the revised manuscript:

"Let us consider a model simulation with an additive random error

$s_1 = o + \varepsilon,$         (36)

where we assume $\mu_{s1} = \mu_o = 0$, and $\sigma_o = \sigma_e$. This simulation in deed gives us the thresholds $NSE_1 = 0, NDE_1 = 0.5, KGE_1 = 0.5$ that distinguish good simulations from bad ones as we have examined in Sect. 2. It is very clear that we cannot improve this simulation since the power of random noise is equal to the power of observations. But this is not true if we measure performances with NSE, NDE, and KGE by constructing a new simulation which is half of $s_1$

$s_2 = 0.5s_1 = 0.5o + 0.5\varepsilon.$           (37)

Calculating its NSE and KGE, we obtain

$NSE_2 = 1 - \frac{\overline{(o - s_2)^2}}{\sigma_o^2} = 1 - \frac{\overline{(0.5o - 0.5\varepsilon)^2}}{\sigma_o^2} = 1 - \frac{0.5\sigma_o^2}{\sigma_o^2} = 0.5,$     (38a)

$NDE_2 = 1 - \frac{\overline{(s_2 - o)^2}}{\overline{s_2^2}} = 1 - \frac{\overline{(0.5\varepsilon - 0.5o)^2}}{\overline{(0.5o + 0.5\varepsilon)^2}} = 1 - \frac{0.5\sigma_o^2}{0.5\sigma_o^2} = 0.,$     (38b)

$KGE_2 = 1 - \sqrt{(\rho - 1)^2 + (\sigma_2/\sigma_o - 1)^2} = 1 - \sqrt{\left(1/\sqrt{2} - 1\right)^2 + \left(1/\sqrt{2} - 1\right)^2} = 2 - \sqrt{2} \approx 0.6$   .

        (38c)

Suddenly, NSE and KGE indicate that $s_2$ is better than $s_1$ considerably, although all we do is just halving $s_1$. In contrast, NDE gives a very different evaluation: $s_2$ is much worse than $s_1$. However, (37) in nature is equivalent to (36), and we should not make any simulation better or worse by just scaling the observations and the random error. This simple example is enough to show that the scientific meaning of the traditional scores like NSE becomes questionable when we introduce multiplicative biases into the error model."

Conclusion: "Its choice is dictated by the fact that at this value the power of noise starts dominating the power of variation of observations." Who dictates it? Why would this interpretation be superior to previous explanations?

Reply 14: We have rewritten this sentence in the revised version:

"The threshold NSE=0 for good/bad model distinction follows naturally from the fact that at this value the power of noise starts dominating the power of variation of observations. The choice of a benchmark model like the observed mean required in the interpretation of such a threshold in the traditional approach is no longer needed in the context of signal processing."

And for this new interpretation in deed forms a sophisticated problem that we have explained in the third reply.

Why would we need to adjust NSE for other error models? It was never intended to be used in conjunction with error models but to yield an easy to interpret performance measure; most authors do not specify an error model; this difference should become clear.

Reply 15: We have explained this sophisticated problem in the Replies 12, 13, 14. In short, we cannot have a universal measure for all cases. The traditional NSE makes sense under the additive error model. However, it is easy to show its failure when other error models are assumed as demonstrated in our example in Reply 13. The situation here is very similar to the case of MSE as examined in Wang and Bovik (2009). Since most authors do not specify an error model explicitly as the reviewer said, we have tried to secure this problem by finding more general versions of NSE and KGE that are applicable in most cases. These efficiencies are described in Section 3 of the manuscript.

Regarding the apparent debate on the meaning of Nash: this needs a reference; did someone else say this or is this your interpretation? I have never heard / seen anyone saying that there is a discussion about what Nash means; there is simply no need to use Nash in other disciplines.

Reply 16: Here we mean that there existed different interpretations for the scientific meaning of NSE. Later in Introduction we have listed three distinct interpretations including reference lists: (1) NSE as the coefficient of determination $R^2$ in linear regression, (2) NSE as a skill score for MSE, and (3) NSE as a multi-criteria score.

**Minor comments:**

Line 9-10: I do not have the original terminology of Nash and Sutcliffe in my head but: NSE is mostly used in model calibration not in forecast quality assessment; in hydrology, a forecast is the prediction of a future state as a function of observed past states; this is a special use of a simulation model. This paper needs reformulating the text in terms of simulations models (input-output models) rather than in terms of forecasts.

Reply: As we have stated in the common answer, the manuscript has been revised in the perspective of model calibration.

Line 23-24: bad use of parentheses

Reply: Thank you for pointing out all typographical and grammatical errors. We have applied proofread for the revised version.

Line 24: what is the benchmark? this is not a term with a universal meaning; what benchmark are you talking about?

Reply: Here the benchmark refers to the benchmark simulation equal to the observed mean in the previous paragraph. However, we have rewritten this sentence in the revised version.

Line 29: as far as I see, the main point of that paper was about the fact that NSE values of different case studies should not be compared; it would have been interesting to reflect on this problem in the present paper.

Reply: As we have stated in the common answer, we have added a discussion on this problem in the revised version.

Line 33-34: this needs a reference; did someone else say this or is this your interpretation? I have never heard / seen anyone saying that there is a discussion about what Nash means;

Reply: Again, here we mean that there existed different interpretations for the scientific meaning of NSE. Later in Introduction we have listed three distinct interpretations including reference lists: (1) NSE as the coefficient of determination $R^2$ in linear regression, (2) NSE as a skill score for MSE, and (3) NSE as a multi-criteria score.

Line 68: strange to give a reference posterior to the KGE suggestion

Reply: We have come back to the original KGE in the revised version.

Line 74-75: what is meant, please revise the sentence

Reply: Thank you. We have rewritten this sentence.

Line 88-89: what does this mean? it is not the error model that excludes forecasts and what is "including the mean forecast"? what is probably meant: With an additive error model, the observational variance is decomposed into the variance of the simulation model plus the variance of the error model; accordingly, the simulation model variance has to be lower than the observational

Reply: Thank you. We have rewritten this sentence.

Line 112: how do you obtain expectation of o^2 = mean^2 + variance; it is probably obvious to most readers, but why not still stay that it follows from the definition of the variance?

Reply: Thank you. We have updated the formula.

Line 157: forecaster are not all men, this is not appropriate in modern scientific writing

Reply: Thank you. We have rewritten this sentence.

Line 187: more than what?

Reply: This means that expression of NSE in terms of $\rho$ better than in terms of RNSR. We have rewritten this sentence.

Line 207: wrong grammar, unclear what the sentence wants to say;

Line 255-256: misleading formulation, here and later: the error model does not exert some sort of selection, it does not exclude; I suggest: "the additive error model implies that the simulations have to a smaller variance than the observations since part of the observation variation is assigned to the error, not to the simulation". besides: this is of course a well-known fact in the hydrological literature dealing with Bayesian model inference (where the model error is inferred along with the model parameters, e.g. paper by Dmitri Kavetski

Reply: Thank you. We have rewritten this sentence.

Line 416: s

Line 420: adding values, what does this mean, adding new samples to the time series? Unclear

Reply: This means that the values of NSE can be easily changed under translations of simulations and observations. We have rewritten this sentence.

Line 422: of

Line 423: ls

Line 428: no, where did you see this? I do not remember having seen this before

Reply: We have cited some researches in Section 2 on this problem:

"This relatively large gap can lead to misjudgement on model performances in practice since similar to NSE, modelers tend to consider KGE=0 as the threshold for good/bad model distinction (Anderson et al., 2017; Fowler et al., 2018; Siqueira et al., 2018; Sutanudjaja et al., 2018; Towner et al., 2019). Thus, all models with KGE between [0,0.5] are wrongly classified to give good performances while they are indeed "bad" models. It is worth noting that Rogelis et al. (2016) assigned the value KGE=0.5 to be the threshold below which simulations are considered to be "poor"."

Line 431-432: what does this sentence mean?

Reply: Thank you. We have rewritten this sentence.

Line 432-433: what is "a distribution between forecasts and observations", what does this mean?

Reply: We mean the joint probability distribution between forecasts and observations.

Line 436-437: what is "it"; what means "it is found to be the traditional"?
Reply: "it" is referred to the generalized NSE.

Line 438: bad writing practice: this use of parentheses is a mis-use of parentheses

Dr. Ding's comments:

Origin of the NSE

This is in response to the plea by the authors for insights on the popular Nash–Sutcliffe model efficiency criterion (NSE) - Lines 79-82.

I would like to share my memory of the origin of an alternate version of NSE. In hindsight, the scientific meaning of NSE boils down to what the term "initial variance" means. This was recently reported in Ding (2018) which may have escaped the attention of the authors.

Using the original notations, the NSE scale is recast from Equation (1) as follows:
$R^2 = (F_0 - F)/F_0,$
$F_0 = \sum(o_i - \mu_o)^2,$
$F = \sum(o_i - f_i)^2,$
in which: $R^2$ is the model efficiency, i.e. NSE; $F_0$ is the "initial variance" of the observations, $o_i$; F is the "residual variance" of the forecasts, $f_i$; subscript i is time; and $\mu_o$ is the mean of the observations.

In the literature, there existed an alternate definition presented by Ding (1974) of the so-called "initial variance", $F_0$. The alternative, symbolized by, say, $F_{0-d}$, appeared four years after Nash and Sutcliffe (1970), not year 1974 as written in the preprint.

I defined $F_{0-d}$ directly from F by replacing the observations by the mean, i.e.
$F_0 = \sum(f_i - \mu_o)^2,$
$F_{0-d}$ thus becomes the sum of squares of the error (SSE) of a model measured against the observed mean flow. This is different from $F_0$ which is simply the variance of the observations.

The origin of an observed-mean-flow benchmark or baseline was embedded as $\mu_o$ in the "initial variance" of both standard and alternate versions. But the alternate version, $F_{0-d}$, shows unambiguously the observed mean flow, a baseline.

The alternative was preserved in Ding (1974, Eqs. (40) & (47)) where the criterion or scale was called, after Nash (1968-69), the model efficiency, $R^2$, same in notation as the coefficient of determination (Line 39). Absent from the paper was the name of Sutcliffe.

Nash was the Journal of Hydrology editor of the 1974 paper of mine, and, as far as I remember, had kept silent on the alternate definition of the "initial variance", a technical term he first coined. In Ding (1974) paper, the raw source, Nash (1968-69), was cited, but not the subsequent one, Nash and Sutcliffe (1970). The alternate efficiency scale could be named after both Nash and Ding as NDE (1974) to differentiate it from the standard NSE (1970) after Nash and Sutcliffe.

Thanks for the opportunity to reflect on the genesis of a classical but imperfect metric in evaluating the performance of hydrologic models. So rudimentary is the observed-mean-flow baseline, and so imprecise the Nash–Sutcliffe efficiency (NSE) scale. But alternatives exist to both the baseline, e.g., Ding (2018), and the standard scale, e.g., Ding (1974). A combination of the two alternatives will lead to an invention of a new metric, "elegant" and "intuitive", both characterizations (Line 79) favoured by the authors of a metric.

Reply: The following efficiency was introduced in Ding (1974) four year after the introduction of NSE in Nash and Sutcliffe (1970)

$$NDE = 1 - \frac{\sum(s_i - o_i)^2}{\sum(s_i - \mu_o)^2} = 1 - \frac{\overline{(s-o)^2}}{\overline{(s-\mu_o)^2}}, \quad (5)$$

where we call this efficiency NDE. This means that instead of regressing observations on simulations as in the case of NSE $o = s + \varepsilon$, NDE regresses simulations on observations $s = o + \varepsilon$ and takes the resulting coefficient of determination as a similarity measure.

Under the additive error model
$$f = o + b + \varepsilon, \quad (6)$$
NDE becomes

$$NDE = 1 - \frac{b^2 + \sigma_e^2}{\sigma_o^2 + b^2 + \sigma_e^2} = \frac{\sigma_o^2}{\sigma_o^2 + b^2 + \sigma_e^2}. \quad (7)$$

This score is surprisingly the square of the correlation efficiency CE that we show in Equation (25) in the old manuscript. We derived CE to illustrate that any score can be constructed from a monotonic function of the correlation coefficient $\rho$, and did not know that CE was indeed proposed in the literature. In the revised manuscript we have added NDE and a citation to Ding (1974).

Thus, NDE is equivalent to NSE, and both measure the noise-to-signal ratio in simulations. However, there is a potential problem here if NDE was chosen instead of NSE in the history. From (5) like NSE we may wrongly conclude that NDE=0 marks the boundary between good and bad simulations. But from (7) this threshold of NDE should be NDE=1/2 when the power of noise starts dominating the power of variation of observations $\sigma_o^2 = b^2 + \sigma_e^2$. Note that in the case of NSE,

both equations similar to (5) and (7) yield the same threshold NSE=0.

Reference

Clark, M. P., Vogel, R. M., Lamontagne, J. R., Mizukami, N., Knoben, W. J. M., Tang, G., et al.: The abuse of popular performance metrics in hydrologic modeling. *Water Resources Research*, *57*, e2020WR029001, doi:10.1029/2020WR029001, 2021.

Ding, J. Y.: Variable unit hydrograph. *Journal of Hydrology*, **22.1-2**, 53-69, 1974.

Knoben, W. J. M., Freer, J. E., and Woods, R. A.: Technical note: Inherent benchmark or not? Comparing Nash–Sutcliffe and Kling–Gupta efficiency scores. *Hydrol. Earth Syst. Sci.*, **23**, 4323–4331, doi:10.5194/hess-23-4323-2019, 2019.

Lamontagne, J. R., Barber, C. A., and Vogel, R. M.: Improved estimators of model performance efficiency for skewed hydrologic data. *Water Resources Res.*, **56**, e2020WR027101, doi:10.1029/2020WR027101, 2020.

Nash, J. E., and Sutcliffe, J. V.: River flow forecasting through conceptual models. Part 1: A discussion of principles. *Journal of Hydrology*, **10(3)**, 282–290, doi:10.1016/0022-1694(70)90255-6, 1970.

Wang, Z., and Bovik, A. C.: Mean squared error: Love it or leave it? A new look at signal fidelity measures. *IEEE Signal Processing Magazine*, **26**(1), 98–117, doi:10.1109/msp.2008.930649, 2009.

---

## Author Response (AR2)

**Response to reviewers**

We would like to thank the two reviewers for their constructive comments to the manuscript. Here are our updates for the technical points that the editors concerned.

Reviewer 1:
I have found an improved paper based on a careful accounting of reviews. In particular I believe this paper may enhance spreading of useful performance metrics such as NSE and KGE, born and well established in the hydrological scientific literature into other fields such as signal processing, in this case.

Reviewer 2:
The revised version is ready for publication. It clearly has added value for the hydrological modelling community since it clearly shows the link between well-known metrics and the signal-to-noise ratio, sheds new light on the interpretation of the well-known metrics and thresholds for good models. Some of the findings have been known previously to the hydrologic model community but are rarely explicitly stated in papers. Many papers show e.g. scatter plots of NSE versus KGE that clearly show that both metrics identify the same best solution (e.g. Yassin et al., Fig. 5, https://hess.copernicus.org/articles/23/3735/2019/ or Knoben et al., Fig. 1, https://hess.copernicus.org/articles/23/4323/2019/) without in detail discussing this.
One point that is missing is the link to Bayesian inference, omni-present in hydrological modelling, with the long standing debate about GLUE, Generalized Likelihood Uncertainty Estimation (e.g. Mantonvan and Todini, 2006). In the Bayesian inference body of literature, there formal assumption about the error model whereas in GLUE, often NSE is used. This link would certainly be useful.

Reply: Thank you for pointing out the link between our study and Bayesian inference. We have added text to highlight this link in the revised manuscript in the beginning of Conclusion:
"This view underlines an important role of the error model between simulations and observations, which is usually implicit in our assumption. Thus, our approach follows Bayesian inference in which an error model is formally defined first, then a goodness-of-fit measure is derived (Mantovan and Todini, 2006; Vrugt et al., 2008). The rational is to avoid the use of NSE as a predefined measure without an explicit error model like in generalized likelihood uncertainty estimation (Beven and Binley, 1992) which has caused a long debate in hydrology community (Mantovan and Todini, 2006; Stedinger et al., 2008)."
Also, a citation to Yassin et al. (2019) has been added to demonstrate that the equivalence between NSE and KGE has been observed in practice.